



# A global climatology of ice nucleating particles at cirrus conditions derived from model simulations with EMAC-MADE3

Christof G. Beer[1], Johannes Hendricks[1], and Mattia Righi[1]

[1]Deutsches Zentrum für Luft- und Raumfahrt (DLR), Institut für Physik der Atmosphäre, Oberpfaffenhofen, Germany

**Correspondence:** Christof Beer (christof.beer@dlr.de)

**Abstract.** Atmospheric aerosols can act as ice nucleating particles (INPs) and thereby influence the formation and the microphysical properties of cirrus clouds resulting in distinct climate modifications. From laboratory experiments several types of aerosol particles have been identified as effective INPs at cirrus conditions. However, the global atmospheric distribution of INPs in the cirrus regime is still highly uncertain as in situ observations are scarce and limited in space and time. To study the

influence of INPs on cirrus clouds and climate on the global scale these particles have been simulated with global chemistry-climate models. Typically, mineral dust and soot particles, which are known to initiate ice nucleation in cirrus clouds, have been considered in these models. In addition, laboratory studies suggest crystalline ammonium sulfate and glassy organic particles as effective INPs in the cirrus regime. However, the representation of these particles in global models is challenging as their phase state, i.e. crystalline or glassy, needs to be simulated. In turn, crystalline ammonium sulfate and glassy organics have

only rarely been considered in global model studies and their impact on the global scale is still uncertain. Here, we present and analyse a global climatology of INPs derived from global model simulations performed with the ECHAM/MESSy Atmospheric Chemistry (EMAC) general circulation model including the aerosol microphysics submodel MADE3 (Modal Aerosol Dynamics model for Europe, adapted for global applications, third generation) coupled to a two-moment cloud microphysical scheme and a parametrization for aerosol-induced ice formation in cirrus clouds. This global INP-climatology comprises mineral dust and soot particles, as well as crystalline ammonium sulfate and glassy organics, including a simplified formulation

of the particle phase state for the latter. By coupling the different INP-types to the microphysical cirrus cloud scheme, their ice nucleation potential at cirrus conditions is analysed, considering possible competition mechanisms between different INPs. The simulated INP concentrations in the range of about 1 to 100 L$^{-1}$ agree well with in situ observations and other global model studies. We show that INP concentrations of glassy organics and crystalline ammonium sulfate are strongly related to the ambient conditions which often inhibit the glassy or crystalline phase, respectively. Our model results suggest that glassy

organic particles probably have only minor influence, as typical INP concentrations are mostly low in the cirrus regime. On the other hand, crystalline ammonium sulfate often shows large INP concentrations, has the potential to influence ice nucleation in cirrus clouds, and should be taken into account in future model applications.





# 1    Introduction

Atmospheric aerosol particles can exert important influences on the global climate system by directly changing the Earth's energy budget via interactions with solar and terrestrial radiation (Boucher et al., 2013; Bellouin et al., 2020). Additionally, aerosol particles can act as cloud condensation nuclei and ice nucleating particles, consequently influencing the formation of cloud droplets and ice crystals and, in turn, leading to additional climate modifications (Boucher et al., 2013; Mülmenstädt and Feingold, 2018). However, these aerosol-cloud interactions are still poorly understood and the subject of ongoing research

activities (Mülmenstädt and Feingold, 2018; Bellouin et al., 2020; Murray et al., 2021). Especially INP-effects on cirrus clouds and their climatic impacts are highly uncertain (e.g. Kärcher, 2017; Kanji et al., 2017).

INPs can initiate heterogeneous nucleation of ice crystals at lower supersaturations with respect to ice compared to the homogeneous freezing of liquid aerosols (Koop et al., 2000; Hoose and Möhler, 2012). This competition mechanism between heterogeneous and homogeneous nucleation for the available supersaturated water vapour crucially depends on the abundance

of INPs and their freezing potential (Kärcher et al., 2006; Gasparini and Lohmann, 2016; McGraw et al., 2020). However, the atmospheric concentration of INPs and their global distribution is still poorly constrained, contributing a large part to the uncertainty in the quantification of the climatic impact of aerosol-cirrus interactions. A global analysis of INP-concentrations is challenging as in situ observations are scarce and mostly targeted the mixed-phase temperature regime (Rogers et al., 2001a; DeMott et al., 2010; Schrod et al., 2017), while measurements at cirrus conditions are very limited. Additionally, measurements

are typically limited in space and time (e.g. Rogers et al., 1998), complicating interpretations on the global scale. In the past, modelling studies applying GCCMs (Global Chemistry Climate Models) have been performed to elucidate the global distribution of different INPs (e.g. Barahona et al., 2010; Hendricks et al., 2011; Penner et al., 2018). However, as the ice nucleating potential of INPs is continually reevaluated in laboratory measurements (Kanji et al., 2017) and new types of INP-species have been suggested to have important influences (e.g. Ladino et al., 2014; Ignatius et al., 2016; Wilbourn et al., 2020),

an updated picture of the global INP-distribution is necessary.

Here we present results from global model simulations of ice nucleating aerosol particles and describe an up-to-date global climatology of INPs, including several major INP-types in the cirrus regime, i.e. mineral dust, soot, crystalline ammonium sulfate and glassy organics. We employ the atmospheric chemistry general circulation model EMAC (ECHAM/MESSy2 Atmospheric Chemistry model; Jöckel et al., 2010) including the MESSy (Modular Earth Submodel System) aerosol microphysics

submodel MADE3 (Modal Aerosol Dynamics model for Europe, adapted for global applications, third generation; Kaiser et al., 2014; 2019). MADE3 is especially suited for the analysis of ice nucleating aerosol processes, as it is able to simulate different particle compositions and mixing states, which influence important ice nucleation properties of the INPs. For instance, insoluble particles with and without relevant soluble coatings can be distinguished in different size ranges. The MADE3 aerosol is coupled to (cirrus) clouds via a two-moment cloud scheme (Kuebbeler et al., 2014) as described in detail by Righi et al. (2020).

This allows to analyse the ice nucleation potential of various INP-types at cirrus conditions, including possible competition mechanisms between different INPs.





Mineral dust and soot particles, which are typically considered as INPs in the cirrus regime (Möhler et al., 2006, 2008; Kulkarni et al., 2016), are represented in EMAC-MADE3 and coupled to the freezing scheme as described in Righi et al. (2020). This also includes the improved representation of mineral dust aerosol by employing an online calculation of wind-driven dust emissions (Tegen et al., 2002) as described in Beer et al. (2020). Furthermore, the model includes an additional tracer for tagging soot particles from specific sources, e.g. from aviation emissions (Righi et al., 2021), to analyse the global distribution of aviation soot INPs.

In addition to mineral dust and soot, other aerosol species have been reported to potentially nucleate ice in the cirrus regime. Several studies described the ice nucleation potential of glassy organic particles (Murray et al., 2010; Ladino et al., 2014; Ignatius et al., 2016; Wagner et al., 2017), often well below the homogeneous freezing threshold. Atmospheric secondary organic aerosol (SOA) particles can exist in a highly viscous, amorphous state, depending on ambient temperature and humidity (Reid et al., 2018). This glassy state was shown to be an essential requirement for a high freezing potential of organic particles (Ignatius et al., 2016). Common biogenic SOA precursors are terpenes, a class of organic compounds emitted by various plants, especially in boreal forest regions, e.g. the monoterpene pinene (Laaksonen et al., 2008). Another typical SOA precursor is isoprene, mainly emitted, for instance, from tropical rainforests (Guenther et al., 1995, 2006). Besides glassy organics, crystalline ammonium sulfate ($(NH_4)_2SO_4$) has been reported as an effective INP at cirrus conditions (Abbatt et al., 2006; Wise et al., 2009; Baustian et al., 2010; Ladino et al., 2014). The phase state of ammonium sulfate particles is crucial for their ice nucleation potential, as only crystalline particles were shown to initiate ice nucleation. Ammonium sulfate particles undergo phase transitions depending on ambient temperature and humidity via a hysteresis process, i.e. the transition from aqueous solution droplets to the solid crystalline phase happens at a characteristic relative humidity, the efflorescence relative humidity (ERH), which is different to that of the converse process. The deliquescence relative humidity (DRH) at which solid crystals dissociate to solution droplets is typically larger than the ERH (Martin, 2000; Martin et al., 2003).

These new types of INPs, i.e. glassy organics and crystalline ammonium sulfate, have only rarely been considered in global modelling studies and their potential for cirrus cloud and climate modifications is still highly uncertain (e.g. Abbatt et al., 2006; Penner et al., 2018; Zhu and Penner, 2020). Here we include glassy organics and crystalline ammonium sulfate as INPs in the framework of EMAC-MADE3 including the representation of their phase state, i.e. glassy or crystalline, respectively, depending on the simulated ambient temperature and relative humidity. These model improvements allow us to investigate the global distributions of a large suite of major INP types in the cirrus regime on a self-consistent climatological basis. The coupling of the applied aerosol submodel with a microphysical two-moment cloud scheme, including the major aerosol-induced cirrus formation pathways as well as their competition, allows to further assess the importance of these INP types in cirrus cloud formation. Based on these modelling capabilities new insights in aerosol-cirrus interaction mechanisms and their global variability are gained in the this study.

The paper is organized as follows. In Sect. 2 we describe the modelling framework EMAC-MADE3 with a focus on the representation of crystalline ammonium sulfate and glassy organics and their corresponding phase characterization. The calculation of number concentrations of potential INPs is described in detail in Sect. 3. The simulated aerosol concentrations of mineral dust, (aviation) soot, ammonium sulfate and glassy organics are shown in Sect. 4. In Sect. 5 we analyse the global





distribution of INP concentrations per species, as well as the concentrations of nucleated ice crystals. The main conclusions of this study are highlighted in Sect. 6.

The work presented in this paper is in parts based on the PhD thesis by C. G. Beer (Beer, 2021) and some of the text appeared similarly therein.

## 2 Model description

### 2.1 EMAC-MADE3 model setup

The EMAC model is a global numerical chemistry and climate simulation system and includes various submodels that describe tropospheric and middle-atmosphere processes. It uses the second version of MESSy to connect multi-institutional computer codes. The core atmospheric model is the ECHAM5 (fifth-generation European Centre Hamburg) general circulation model (Roeckner et al., 2006). In this work we apply EMAC (ECHAM5 version 5.3.02, MESSy version 2.55) in the T63L31 configuration with spherical truncation of T63 (corresponding to a horizontal resolution of about $1.9° \times 1.9°$) and 31 non-equidistant vertical layers from the surface to $10 \, \text{hPa}$. The simulated time period covers the years 2009 to 2019, while the year 2009 is used as a spin-up and excluded from the evaluation. All simulations presented here are performed in nudged mode, i.e. model meteorology (temperature, winds and logarithm of the surface pressure) is relaxed towards the ERA5 reanalysis data (Hersbach et al., 2020) for the same time period.

The aerosol microphysics submodel MADE3 simulates different aerosol species in nine log-normal modes that represent different particle sizes and mixing states. A detailed description of MADE3 and its application and evaluation as part of EMAC can be found in Kaiser et al. (2014, 2019).

In this study, EMAC-MADE3 is employed in a coupled configuration which includes a two-moment cloud microphysical scheme based on Kuebbeler et al. (2014), employing a parametrization for aerosol-driven ice formation in cirrus clouds following Kärcher et al. (2006). The Kärcher et al. (2006) scheme considers the competition between various ice formation mechanisms for the available supersaturated water vapour, i.e. homogeneous freezing of solution droplets, deposition and immersion nucleation induced by INPs, and the growth of preexisting ice crystals. In each of the heterogeneous freezing modes the ice-nucleation properties of the INPs are represented by two parameters, namely the active fraction ($f_{\text{act}}$) of ice nucleating particles, which actually lead to the formation of ice crystals, and the critical supersaturation ratio with respect to ice ($S_c$), at which the freezing process is initiated. The model setup has been extensively tuned and evaluated with respect to various cloud and radiation variables by Righi et al. (2020) with further model improvements described in Righi et al. (2021).

In this study, ice nucleation induced by mineral dust (DU), black carbon (BC, in the form of soot particles) from all sources except aviation, BC from aviation (BCair), glassy organic particles (glPOM), and crystalline ammonium sulfate (AmSu) is considered. A summary of the freezing properties of these INPs is presented in Table 1. For ammonium sulfate and organic INPs only a few studies exist that consider specific phase states of these INPs, i.e. particles in a crystalline or glassy state. In this work, the corresponding freezing properties for crystalline ammonium sulfate and glassy organics are assumed according to Ladino et al. (2014) and Ignatius et al. (2016), respectively. Dust immersion freezing was also only rarely investigated in





**Table 1.** Freezing properties of ice nucleating particles in the cirrus regime assumed in this study, i.e. critical supersaturation $S_c$ and activated fraction $f_{\mathrm{act}}$ at the freezing onset. $S_i$ is the supersaturation with respect to ice. In addition to onset $f_{\mathrm{act}}$, values for $f_{\mathrm{act}}$ at about $S_i = 1.4$ are used in the analysis.

| Freezing mode | | $S_c$ | $f_{\mathrm{act}}$ at onset $S_c$ | $f_{\mathrm{act}}$ at $S_i = 1.4$ | Reference |
|---|---|---|---|---|---|
| DU deposition | $T \leq 220\,\mathrm{K}$ | 1.10 | $\exp[2\,(S_i - S_c)] - 1$ | 0.822 | Möhler et al. (2006) |
| | $T > 220\,\mathrm{K}$ | 1.20 | $\exp[0.5\,(S_i - S_c)] - 1$ | 0.105 | |
| AmSu | | 1.25 | 0.001 | 0.02 | Ladino et al. (2014) |
| glPOM | | 1.30 | 0.001 | 0.08 | Ignatius et al. (2016) |
| DU immersion | | 1.35 | 0.01 | 0.1 | Kulkarni et al. (2014) |
| BC aviation | | 1.40 | 0.001 | 0.001 | e.g. Kulkarni et al. (2016) |
| BC other sources | | 1.40 | 0.001 | 0.001 | e.g. Kulkarni et al. (2016) |

laboratory experiments. In the original Kuebbeler et al. (2014) scheme, dust immersion freezing with $S_c = 1.3$ was assumed according to Möhler et al. (2008). Here a slightly larger value of $S_c = 1.35$ is assumed according to Kulkarni et al. (2014), where the mixed-phase regime at $T = 238\,\mathrm{K}$ is considered, with sulfuric acid coatings around dust particles. Despite the mixed-phase temperature, this value is used as a more conservative assumption in the cirrus regime, as Möhler et al. (2008) only considered dust particles with organic coatings at very low temperatures ($205\,\mathrm{K} < T < 210\,\mathrm{K}$). For DU deposition freezing,

$f_{\mathrm{act}}$ is calculated according to Kuebbeler et al. (2014), depending on the supersaturation with respect to ice and the temperature. There exist a large number of different studies investigating the freezing of soot particles from a variety of different sources. For example, Koehler et al. (2009) analysed different soot types, including soot resulting from burning aviation kerosene, and observed ice nucleation below the homogeneous freezing threshold. Kanji et al. (2011) reported ice nucleation on graphite soot at supersaturations $S_c$ between 1.3 and 1.5, while Chou et al. (2013) and Kulkarni et al. (2016) measured $S_c$ around 1.4 for

fresh and aged diesel soot particles. Recent studies investigated BC ice nucleation at cirrus temperatures and attributed this to the pore condensation and freezing mechanism rather than deposition freezing (e.g., Marcolli, 2017; Mahrt et al., 2018; David et al., 2019; Mahrt et al., 2020). This was shown to be related to the cloud processing of soot particles (e.g. in contrails) and the resulting enhancement of their freezing potential by lowering $S_c$. In this study a value of $S_c = 1.4$ for BC (and aviation BC) ice nucleation is assumed in accordance with the wide range of results from measurements.

The EMAC-MADE3 setup applied here is largely based on the setup described in Righi et al. (2021). We use the recent CMIP6 (Coupled Model Intercomparison Project, phase 6) emission inventory for anthropogenic and biomass burning emissions of aerosols and aerosol precursor species (van Marle et al., 2017; Hoesly et al., 2018) for the year 2014, to provide an emission setup close to present day conditions. Mineral dust emissions are calculated according to the online emission scheme of Tegen et al. (2002), as described and evaluated in Beer et al. (2020). We employ EMAC in the T63L31 resolution, in contrast





to the T42L41 resolution in Righi et al. (2021), as this was shown to improve the simulated aerosol concentrations, especially for mineral dust, in the upper troposphere (Beer et al., 2020).

For this study, further developments of the model system were performed concerning the implementation of additional aerosol species that can act as ice nucleating particles, i.e. glassy organics and crystalline ammonium sulfate. The model representation of these new particle types is described in the following (Sect. 3.1–Sect. 3.3). The calculations of the num-

ber concentrations of potential INPs, i.e. glassy organics, crystalline ammonium sulfate, mineral dust, and black carbon are presented in Sect. 3.4.

## 3   Implementation of additional ice nucleating aerosol species

### 3.1   Black carbon from aviation

As described in Righi et al. (2021), an additional black carbon tracer for tagging soot emissions from the aviation sector (BCair)

is used. BCair is implemented as an additional MADE3 aerosol tracer and is distributed in the same six modes as the standard BC tracers, namely insoluble and mixed Aitken, accumulation, and coarse mode. Compared to the standard BC tracer, BCair has the same physical properties and is subject to the same processes in the model.

### 3.2   Glassy organics

Similar to BCair, a MADE3 tracer for glassy organic particles (glPOM) was included in addition to the standard POM (Par-

ticulate Organic Matter) tracer, considering emissions of natural SOA precursors (e.g. terpenes) taken from Guenther et al. (1995). The precursor gases include isoprene, monoterpenes and other volatile organic compounds, which are mainly emitted from tropical woodlands, especially rainforests. In analogy to Kaiser et al. (2019), it is assumed that natural terpenes are transformed to SOA with a constant yield of 15 %, following Dentener et al. (2006). The resulting SOA species are assumed to irreversibly condense as particulate organic matter on preexisting aerosol particles. To track SOA from natural terpenes the

condensed mass is assigned to the glPOM tracer.

Importantly, atmospheric SOA particles can be transformed into an amorphous, glassy state, with extremely high viscosities $\eta > 10^{12} \, \mathrm{Pa \, s}$ (Koop et al., 2011), facilitating their ice nucleating potential. This glass transition critically depends on the atmospheric conditions. If the ambient temperature decreases below a certain threshold temperature, namely the glass transition temperature ($T_g$), a liquid solution particle vitrifies and is transformed into a semi-solid, glassy state (Reid et al., 2018).

For increasing temperatures, the reverse process, i.e. the glass-to-liquid transition, occurs at the same specific temperature. $T_g$ depends on the composition of the organic compound and increases with decreasing relative humidity (RH). For small humidities (RH $< 10 \%$), Koop et al. (2011) describe a $T_g$ for a SOA estimate of around $270 \, \mathrm{K}$. Here, we use three different formulations for $T_g$ depending on RH, assuming citric acid ($T_g^{(1)}$), glucose ($T_g^{(2)}$), and sucrose ($T_g^{(3)}$), which have been shown to be good proxies for atmospheric SOA (Baustian et al., 2013). $T_g$ in units of Kelvin is given in Baustian et al. (2013) for





relative humidities RH between $0\,\%$ and $100\,\%$ for the three SOA proxies:

$$T_g^{(1)}(\mathrm{RH}) = 277.14 - 0.33\,\mathrm{RH} - 0.01\,\mathrm{RH}^2 \tag{1a}$$

$$T_g^{(2)}(\mathrm{RH}) = 293.26 + 0.12\,\mathrm{RH} - 0.016\,\mathrm{RH}^2 \tag{1b}$$

$$T_g^{(3)}(\mathrm{RH}) = 333.94 - 0.3\,\mathrm{RH} - 0.017\,\mathrm{RH}^2 \tag{1c}$$

The three different $T_g(\mathrm{RH})$ curves are shown in Fig. S1 in the Supplement. The reference case for the model simulations is
$T_g^{(2)}(\mathrm{RH})$, the other two estimates represent a lower and an upper boundary for $T_g$. For the glPOM tracer we only consider those cases with $T < T_g$.

### 3.3   Ammonium sulfate

In contrast to glPOM, crystalline ammonium sulftate $((\mathrm{NH_4})_2\mathrm{SO_4})$ requires the representation of a dedicated phase transition, including a hysteresis process depending on the history of the relative humidity. Crystallization of ammonium sulfate occurs
at a lower relative humidity (efflorescence, ERH) than the transition from crystals to aqueous particles (deliquescence, DRH). As no phase state characterization is possible for MADE3 aerosol tracers, a different approach is applied here. Crystalline ammonium sulfate is realized as a passive aerosol tracer (AmSu) using the EMAC submodel PTRAC for prognostic or passive tracers (Jöckel et al., 2008). In PTRAC, particle radius, density and geometric standard deviation are fixed at predefined constant values for each prognostic tracer. As the number concentration of INPs is the central quantity for ice nucleation
processes, AmSu is implemented as a number density tracer, and is calculated from ammonium $(\mathrm{NH_4})$ and sulfate $(\mathrm{SO_4})$ input masses from MADE3 assuming characteristic particle sizes. In order to realize different assumptions for the representation of ammonium sulfate and to assess related uncertainties, three different PTRAC tracers are defined:

**AmSu$_{\mathrm{mixS}}$**: Crystalline ammonium sulfate particles with inclusions of other components and $100\,\mathrm{nm}$ particle radius. This represents typical MADE3 accumulation mode sizes for polluted air. The formulation of phase transitions is described
below. For this tracer, only grid boxes where the fraction of $\mathrm{NH_4}$ and $\mathrm{SO_4}$ dominates the total grid box dry mass $(> 50\,\%)$ are considered.

**AmSu$_{\mathrm{mixL}}$**: Similar to AmSu$_{\mathrm{mixS}}$, but considering crystalline ammonium sulfate particles with inclusions of other components and $250\,\mathrm{nm}$ particle radius. This represents larger accumulation mode particles. Again, only grid boxes dominated by $\mathrm{NH_4}$ and $\mathrm{SO_4}$ are considered.

**AmSu$_{\mathrm{ext}}$**: Externally mixed crystalline ammonium sulfate (without inclusions) and $100\,\mathrm{nm}$ particle radius. This represents a control estimate for the crystalline ammonium sulfate particle number concentration, independent of the MADE3 representation of the mixing state. Here, all available ammonium and sulfate mass in the accumulation mode is used, without the restriction to $\mathrm{NH_4}$ and $\mathrm{SO_4}$ dominated particles.

These AmSu tracers are calculated according to $\mathrm{NH_4}$ and $\mathrm{SO_4}$ input masses from MADE3 soluble and mixed accumulation
modes. As AmSu$_{\mathrm{mixS}}$ and AmSu$_{\mathrm{mixL}}$ have different particle diameters, the total soluble and mixed accumulation mode masses have to be separated into two fractions. This is done using the intersection radius $R_{\mathrm{inter}} = (R_{\mathrm{AmSu_{mixS}}} + R_{\mathrm{AmSu_{mixL}}})\,/\,2 = 175\,\mathrm{nm}$, in analogy to the MADE3 mode renaming operation described in Kaiser et al. (2014). The accumulation modes are





most relevant for ice nucleation induced by crystalline ammonium sulfate (insoluble modes carry too less $NH_4$ and $SO_4$).
Aitken mode particles are too small to be efficient ice nuclei (e.g. Kanji et al., 2017), while coarse modes have only few par-

ticles. However, as large coarse mode particles can be highly efficient ice nuclei, two additional tracers for internally and ex-
ternally mixed coarse ammonium sulfate ($AmSu_{mix,coa}$ and $AmSu_{ext,coa}$) are defined similarly to the AmSu tracers described
above, but with 1.75 µm and 1.0 µm particle radii, respectively. These sizes are chosen similar to the AmSu accumulation mode
tracers but are one order of magnitude larger, representing typical coarse mode sizes[1]. The specific density of AmSu tracers is
set to 1770 $kg\,m^{-3}$ (Rumble, 2004). The geometric standard deviations are defined in analogy to the MADE3 accumulation

and coarse modes as $\sigma_g = 2.0$ for the accumulation mode, and $\sigma_g = 2.2$ for the coarse mode, respectively (Kaiser et al., 2019).
The calculations of AmSu tracers consider phase transitions according to the relative humidity, as well as changes of tracer
concentrations due to removal processes in clouds and precipitation (dry deposition of AmSu is represented similarly to other
aerosol tracers). The subroutines dealing with these calculations are part of the MADE3 submodel and are described in detail
in Appendix A.

## 3.4 Calculation of number concentrations of potential INPs

In order to realize ice nucleation induced by aerosol particles in the model, the corresponding aerosol tracers need to be coupled
to the cloud parametrizations in the EMAC CLOUD module. There, the number concentrations of potential INPs are calculated
for the different ice formation modes in the mixed-phase and the cirrus regime according to the procedure described in Righi
et al. (2020). Potential INP numbers are multiplied with the respective ice-active fractions $f_{act}$ to be used as input for the

cirrus cloud parametrization. Here, the calculations described in Righi et al. (2020) are expanded to include the additional INPs
considered in this study, i.e. glPOM, AmSu and BCair. The calculation of INP concentrations available for freezing events is
calculated for every MADE3 mode as described in detail in Appendix B.

For the freezing of crystalline ammonium sulfate, the sum of $AmSu_{mixS}$ and $AmSu_{mixL}$ number concentrations is consid-
ered, as these tracers provide a more detailed and realistic representation compared to $AmSu_{ext}$. As the cloud scheme considers

additional increases in relative humidity due to subgrid-scale updrafts, AmSu freezing is only considered if $S_c < DRH$, as the
particles would otherwise not be in the crystalline phase. For this condition the supersaturation with respect to ice is converted
to the value with respect to liquid water using the formulations described by Murphy and Koop (2005), depending on the ratio
of the vapour pressures of liquid water and ice, respectively. As the the cloud scheme requires information about AmSu number
concentrations per MADE3 mode, AmSu numbers are separated according to $NH_4$ and $SO_4$ masses in MADE3 soluble and

mixed modes. Additionally, possible inclusions in AmSu particles (e.g. DU) have to be subtracted from the corresponding
concentration in other freezing modes.

Generally, in every mode where glPOM is present, the number of glPOM INPs is calculated first and subtracted from the
number in the other freezing modes, as glassy organics are assumed to form a shell around other particles (e.g. Smith et al.,
2012, 2013; Schill et al., 2014; Saukko et al., 2015). This shell is then the relevant part of the particle for the ice nucleation

---

[1] For simplicity, only one mixed coarse mode AmSu tracer is considered, with a size in the middle between 1.0 µm and 2.5 µm, similar to the accumulation
mode mixed tracers but one order of magnitude larger.





processes. An alternative formulation for glPOM freezing, considering glPOM INPs only for grid boxes where the glPOM mass-fraction (with respect to the total mass in the mode) exceeds certain thresholds (e.g. 0.3 or 0.5), resulted in negligible number concentrations of potential glPOM INPs.

## 4 Atmospheric distribution of aerosols acting as ice nucleating particles

In this section we present the atmospheric distributions of the aerosol species that act as ice nucleating particles in the model, i.e.
mineral dust, black carbon, glassy organic particles, and crystalline ammonium sulfate. These simulated aerosol concentrations are the input for the ice nucleation scheme where the actual number concentration of the potential INPs is calculated (see Sect. 3.4).

### 4.1 Mineral dust and black carbon

Figure 1 shows the dust, BC, and aviation BC mass concentrations as simulated by EMAC-MADE3 at $300\,\mathrm{hPa}$ (Fig. 1a, c,
e), and the zonal-mean vertical distribution (Fig. 1b, d, f). As mineral dust is a primary aerosol, dust concentrations in the atmosphere are strongly connected to its emission regions (most prominently the Sahara and Arabian deserts). Concentrations of up to $0.5\,\mathrm{\mu g\,m^{-3}}$ are reached in the upper troposphere over parts of northern Africa, with a strong vertical gradient towards largest concentrations close to the surface. A more detailed description and evaluation of mineral dust in the EMAC-MADE3 model is presented in Beer et al. (2020).
Black carbon mass concentrations simulated with EMAC-MADE3 are shown in Fig. 1c, d. The highest BC concentrations occur in the Northern Hemisphere, where main sources of anthropogenic soot particles, produced from incomplete combustion of fossil fuels, are situated. Additionally, biomass burning is a source of atmospheric BC in remote regions of the Southern Hemisphere (e.g. in Africa and South America). Compared to dust mass concentrations, BC mass concentrations are much lower with peak values of up to $0.01\,\mathrm{\mu g\,m^{-3}}$ in the upper troposphere.
Aviation BC is mostly concentrated in the Northern Hemisphere where the majority of the global air traffic is located. Largest concentrations in the upper troposphere at $300\,\mathrm{hPa}$ reach values of about $10^{-4}\,\mathrm{\mu g\,m^{-3}}$ (Fig. 1e) and the vertical profile shows hotspots for BCair concentrations at the typical flight altitudes (between 200 and $300\,\mathrm{hPa}$) and close to the surface, due to BC emitted in the vicinity of airports (Fig. 1f). Further details on aviation soot in the EMAC-MADE3 model can be found in Righi et al. (2021).

### 4.2 Glassy organics

In this section, model results concerning the atmospheric dispersion of glassy organic particles are presented. The newly implemented MADE3 tracer glPOM represents organic particles formed from the condensation of natural precursors of organic aerosols, e.g. terpenes (see Sect. 3.2). Emissions occur mainly in regions with strong biogenic activity, e.g. tropical rainforests. Only high-viscosity, glassy particles are considered according to the phase separation depending on the ambient temperature
and the glass transition temperature $T_g$.





**Figure 1.** Global distribution of mass concentrations of mineral dust (DU; a, b), black carbon (BC; c, d), and black carbon from aviation emissions (BCair; e, f) in units of µg m$^{-3}$, simulated with EMAC-MADE3 considering the multi-year average over the simulated period (2010–2019). Panels (a), (c), and (e) show the global distribution at the 300 hPa pressure level, panels (b), (d), and (e) show zonal means. Note the different scales in each panel.

The glass transition temperature is strongly influenced by the relative humidity according to Eq. (1), with $T_g$ increasing with decreasing RH. Only for $T < T_g$ glassy particles can occur. This condition is only rarely fulfilled near the surface. The highest mass concentrations (about $0.01\ \mu\mathrm{g\,m^{-3}}$) are reached around the 600 hPa niveau (Fig. 2a, b). There, the low ambient





temperatures favour the occurrence of the glassy phase. Additionally, dry environments with low RH and therefore high $T_g$
lead to favourable conditions for glassy particles, e.g. the northern and southern midlatitudes around $600\,\mathrm{hPa}$ in addition to
midlatitudes and polar regions in the upper troposphere above $200\,\mathrm{hPa}$ (Fig. 2c). This is in agreement with measurements of
glassy SOA particles, which were observed to exist in an amorphous solid state at cirrus temperatures (e.g. Järvinen et al.,
2016), while mostly liquid SOA particles were observed in humid tropical regions (e.g. Bateman et al., 2015).

The ratio $T_g/T$ can be used as an indicator of the particle phase state (Fig. 2d). For $T_g/T \geq 1$ the particle behaves like a solid,
while $T_g/T < 1$ indicates a semi-solid or liquid state (Shiraiwa et al., 2017). A comparison of the ratio $T_g/T$ with model results
from Shiraiwa et al. (2017), also employing the global model EMAC, shows a good agreement. Above $500\,\mathrm{hPa}$ almost all SOA
particles are transformed into a glassy solid state with $T_g/T$ values above 1. Instead of the simple calculation depending on the
relative humidity adopted here, Shiraiwa et al. (2017) employ the organic aerosol submodule ORACLE (Tsimpidi et al., 2014)
to simulate the phase state of atmospheric SOA. Our implementation of glPOM in the EMAC-MADE3 model represents a
simplified approach to derive a first order estimate of the highly uncertain climate effects regarding cirrus cloud modifications
due to glassy organic INPs. In a sensitivity experiment we analyse the effect of using different formulations for $T_g(\mathrm{RH})$ (see
Sect. 3.2 and Fig. S2 in the Supplement). Increased $T_g$ values lead to larger glPOM concentrations mainly at lower altitudes (up
to $500\,\mathrm{hPa}$), as conditions favourable for glassy particles are more frequently fulfilled. Above $300\,\mathrm{hPa}$ only small differences
in glPOM concentrations are visible.

## 4.3 Crystalline ammonium sulfate

In this section, model results concerning the newly implemented ammonium sulfate tracer (AmSu) are presented. A detailed
description of related calculations and parametrizations of the life cycle of atmospheric ammonium sulfate particles in EMAC-
MADE3 is given in Sect. 3.3. AmSu is implemented as a number density tracer and includes a formulation of phase transitions
depending on the ambient relative humidity, i.e. only the crystalline phase is represented by this tracer. To analyse the sensitivity
of modelled AmSu to different assumptions for its size and composition, different representations of AmSu tracers are used
(see Sect. 3.3). $\mathrm{AmSu_{mixS}}$ and $\mathrm{AmSu_{mixL}}$ consider crystalline ammonium sulfate with smaller and larger particle diameters
internally mixed with other components, while $\mathrm{AmSu_{ext}}$ assumes externally mixed pure ammonium sulfate crystals. Number
concentrations of these three AmSu tracers near the surface, at the $300\,\mathrm{hPa}$ level, and as zonal-mean vertical distribution are
shown in Fig. 3. These results represent averages over the whole simulated time period, which includes cases where ammonium
sulfate occurs in its crystalline form but also cases where the environmental conditions exclude the presence of AmSu crystals
and the respective AmSu tracer concentration is therefore zero.

The global patterns are similar for all three AmSu tracers. High number concentrations are simulated at ground level over the
continents and in large parts of the middle and upper troposphere. $\mathrm{AmSu_{mixL}}$ shows the lowest concentration values, as these
particles have a larger diameter, which leads to lower number concentrations in the mass-to-number conversion of the aerosol
mass (Sect. 3.3 and Eq. A6). Largest number concentrations are found for $\mathrm{AmSu_{ext}}$ (above $50\,\mathrm{cm}^{-3}$ near the surface). The
sum of $\mathrm{AmSu_{mixS}}$ and $\mathrm{AmSu_{mixL}}$ concentrations is similar to that of $\mathrm{AmSu_{ext}}$ indicating a low sensitivity to the assumption
of internally or externally mixed particles.





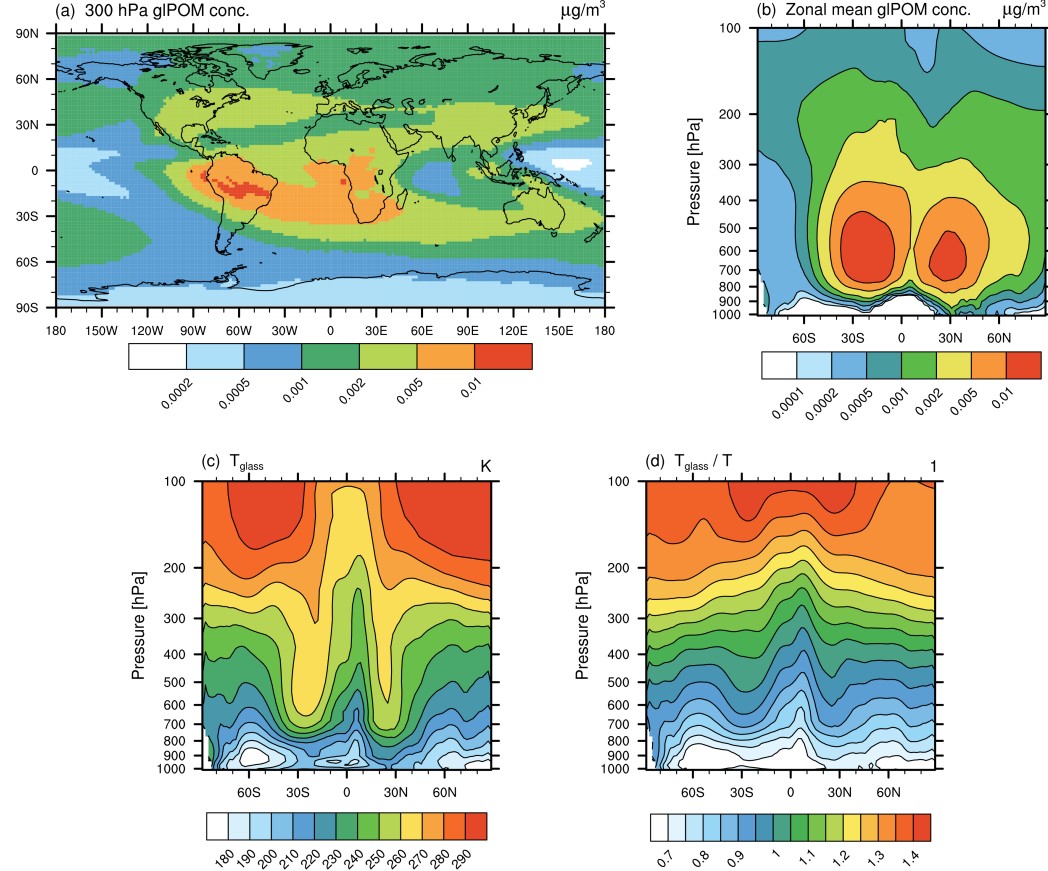

**Figure 2.** Global distribution of glassy organics (glPOM) at the 300 hPa pressure level (a), and as zonal mean (b) in units of $\mu g\,m^{-3}$, as simulated with EMAC-MADE3. Panels (c) and (d) show the zonally averaged glass transition temperature $T_g$ in units of K and the ratio $T_g/T$, respectively. All panels consider the multi-year average over the simulated period (2010–2019). Note the different scales in each panel.

The global dispersion of crystalline ammonium sulfate is strongly related to concentrations of $NH_4$ and $SO_4$ from which it is formed (see Fig. S3 in the Supplement). Sources of aerosol sulfate and ammonium are predominantly of anthropogenic origin, e.g. the combustion of sulfur-containing fossil fuels (like coal in power plants or bunker fuels in shipping) or the use of ammoniacal fertilizers, respectively, and are mostly situated on the Northern Hemisphere (e.g. Feng et al., 2020).

The global distribution of simulated AmSu concentrations is in good agreement with results from other model studies. Wang et al. (2008) analysed the distribution of solid and aqueous sulfate aerosols employing the GEOS-Chem chemical transport model (Martin et al., 2004; Park et al., 2004) and simulated a similar global distribution pattern of solid ammonium sulfate particles as shown in Fig. 3. Penner et al. (2018) used the CAM/IMPACT atmospheric model to simulate the aerosol effect on





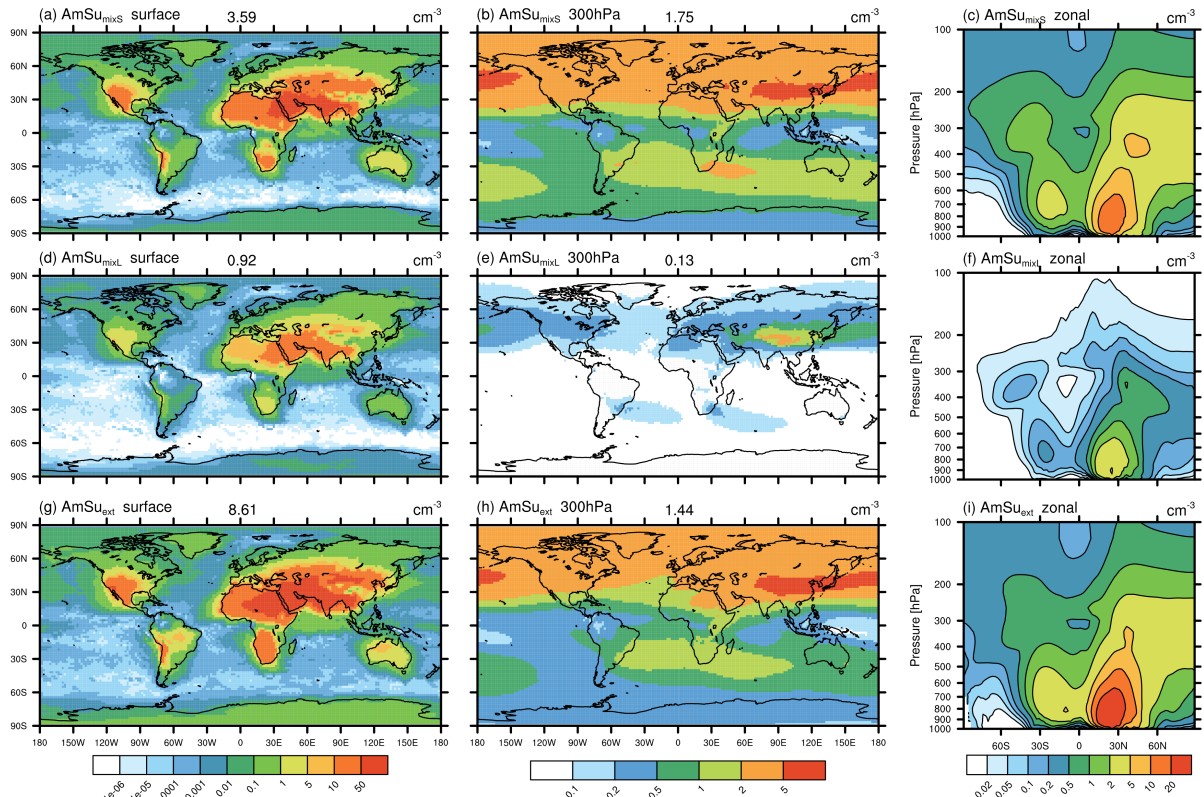

**Figure 3.** Global distribution of crystalline ammonium sulfate (AmSu) simulated with EMAC-MADE3, considering the multi-year average over the simulated period (2010–2019). Number concentrations in units of $cm^{-3}$ are shown for different AmSu tracers, i.e $AmSu_{mixS}$ (a, b, c), $AmSu_{mixL}$ (d, e, f), and $AmSu_{ext}$ (g, h, i), respectively. Number concentrations are shown near the surface (a, d, g), at the 300 hPa level (b, e, h), and as zonal-means (c, f, i). Numbers above the map plots represent global averages at that respective pressure level. Note the different scales in each column.

cirrus clouds, including a representation for solid ammonium sulfate in their model and found also similar global distribution patterns of ammonium sulfate particles compared with the ones presented here.

Importantly, the ambient environmental conditions have to fulfill the requirements for ammonium sulfate to reside in the solid phase. Only if the ambient RH is below the efflorescence relative humidity crystals can form. Zonal-mean profiles of
RH and ERH are shown in Fig. 4d, e. These quantities are variable in time, i.e. the analysis of multi-year means allows only for rough estimates of the AmSu phase state. ERH is calculated depending on the ammonium-to-sulfate ratio according to Eq. A4 (see Appendix A for details). Upon rehydration ammonium sulfate crystals remain solid until the deliquescence relative humidity (DRH) is reached, which is 80 % for ammonium sulfate (Martin, 2000).

Figures 4a-c show the simulated solid fractions of ammonium sulfate. Solid fractions increase with increasing altitude and
show the largest values (close to 100 %) in the upper troposphere of the Northern Hemisphere. As crystallization depends on the





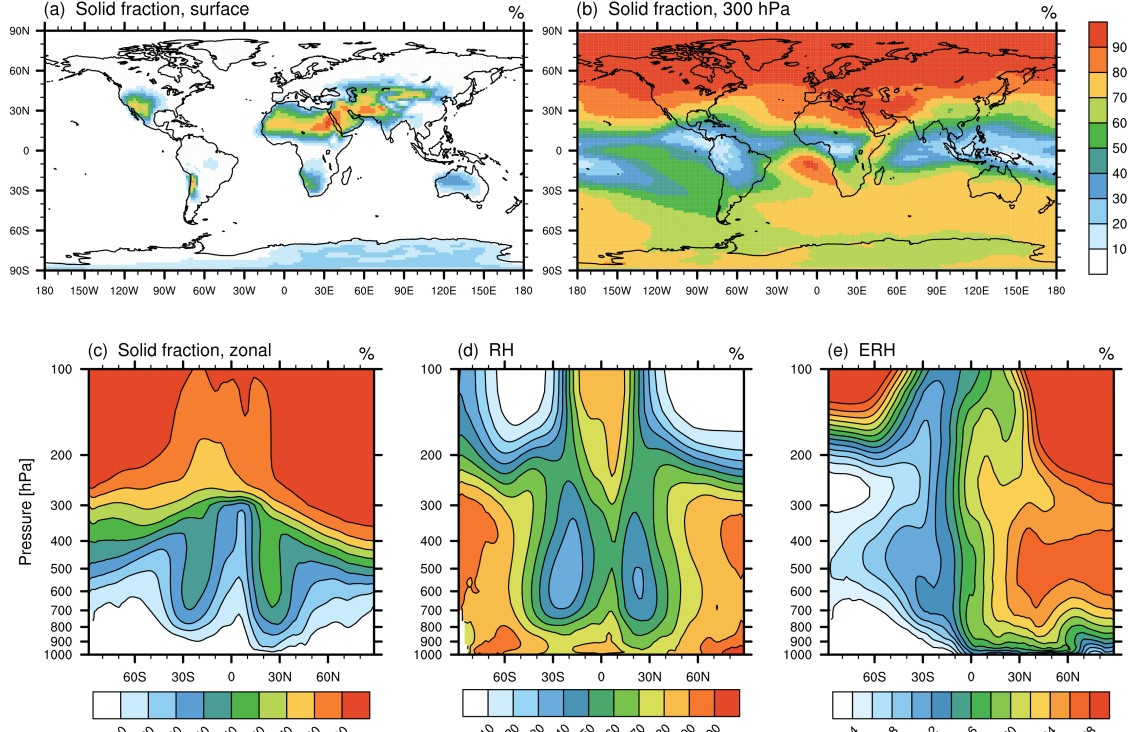

**Figure 4.** Modelled ammonium sulfate solid fractions in EMAC-MADE3, considering the multi-year average over the simulated period (2010–2019). (a-c) Fraction of solid ammonium sulfate particles near the surface, at 300 hPa, and as zonal-mean vertical distribution, (d) zonal-mean relative humidity, (e) zonal-mean efflorescence relative humidity. All quantities are dimensionless and shown in %. Note the different scales in panels (d) and (e).

ambient relative humidity and the efflorescence relative humidity (see Sect. 3.3), large solid fractions occur in regions with low RH and high ERH (see Fig. 4d, e). The simulated solid fractions shown in Fig. 4 are in good quantitative agreement with results from Wang et al. (2008) and also with Colberg et al. (2003), where a Lagrangian model is employed that calculates trajectories from ECMWF (European Centre for Medium-Range Weather Forecasts) analyses and takes the deliquescence/efflorescence

hysteresis of ammonium sulfate into account.

# 5 Climatology of ice nucleating particles

In this section the global distribution of the different INPs simulated with EMAC-MADE3 is presented. Additionally, results concerning the number of pristine, i.e. newly formed ice crystals from heterogeneous freezing, as well as ice water content (i.e. ice mass per unit mass of air) per heterogeneous freezing mode are shown.





Global distributions of number concentrations of the INPs considered in this study (DU, BC, AmSu, glPOM, BCair) in the cirrus regime are depicted in Fig. 5 in terms of vertically averaged multi-annual means over the years 2010–2019. For this comparison the number concentrations of potential INPs (calculated as described in Appendix B) are multiplied by the ice-active fraction $f_{\mathrm{act}}$ provided by measurements (see Table 1), to obtain the actual concentrations of INPs. Here, values of $f_{\mathrm{act}}$ at a supersaturation ratio with respect to ice of $S_i = 1.4$ are chosen. At such a high $S_i$ value near the homogeneous freezing

threshold, all different INP-types are able to nucleate ice. This facilitates a direct comparison of INP number concentrations, as possible biases due to different freezing efficiencies are excluded. For this analysis only grid boxes inside cirrus clouds are considered, selected according to thresholds for simulated ambient temperature ($T < 238$ K) and ice water content (IWC $>$ $0.5\,\mathrm{mg\,kg^{-1}}$), using the original model output frequency of $11\,\mathrm{h}$. The global distribution of the average cirrus cloud occurrence frequency calculated according to these threshold values is shown in Fig. S4 in the Supplement. An additional selection criterion

is employed to filter out those cases where the crystal size exceeds a certain threshold ($R_{\mathrm{ice}} > 1000\,\mu\mathrm{m}$), as such large crystals readily sediment and are removed from the cirrus niveau. This threshold value is chosen according to the analysis of crystal sizes of newly formed ice crystals (see Fig. S5 in the Supplement) to include most of the distribution of simulated $R_{\mathrm{ice}}$ values, while excluding very large crystals.

    Peak INP number concentrations per species in Fig. 5 reach values of 50 to $100\,\mathrm{L^{-1}}$ for most INP types, while total INP num-

ber concentrations show values of up to $200\,\mathrm{L^{-1}}$. Mineral dust INPs are mostly concentrated near strong dust emission regions (e.g. the Sahara or Arabian Desert), but also in regions of enhanced dust transport (e.g. over the Atlantic Ocean). The INP-types BC, BCair and AmSu show a strong hemispheric gradient with high concentrations on the Northern Hemisphere, as these INPs are strongly related to anthropogenic activities (e.g. combustion of fossil fuels). For the case of BCair the main aviation flight corridors are clearly visible, which are dominated by air traffic over the Northern Atlantic Ocean. BCair number concentrations

of up to $2\,\mathrm{L^{-1}}$ are generally smaller compared to other INP-types. However, this represents a conservative assumption and BCair numbers could be larger, as aircraft soot could possibly be preactivated via processing in contrails (e.g., Mahrt et al., 2018; David et al., 2019; Nichman et al., 2019), while here the activated fraction of BCair ($f_{\mathrm{act}} = 0.003$) is chosen according to the value for BC. Glassy organic INPs are more homogeneously distributed over the Northern and Southern Hemisphere with highest concentrations over regions of strong biogenic activity (e.g. tropical rainforests) with number concentrations mostly

around $5\,\mathrm{L^{-1}}$, distinctly smaller compared to most other INP-types. As stated earlier, glassy organics as well as crystalline ammonium sulfate INPs occur only under specific conditions, as their phase state depends on the ambient temperature and humidity. Differences of the global distribution of INPs as seen in Fig. 5 compared to the respective aerosol concentrations shown in the previous sections are a result of the selection of cirrus conditions (i.e. $T < 238\,\mathrm{K}$, IWC $> 0.5\,\mathrm{mg\,kg^{-1}}$), but can also be related to the assumptions for the calculation of INP number concentrations as described in Sect. 3.4 and in Appendix B.

A direct comparison of model results with in situ observations of INP number concentrations in cirrus clouds is challenging, as most measurements were performed at lower altitudes and focused on mixed-phase cloud temperatures. Moreover, in situ measurements of collected particle samples were often performed using diffusion chambers (e.g., Rogers et al., 2001b), where temperatures and supersaturations can be directly controlled but may not correspond to the actual ambient conditions at sample collection, e.g. temperatures below the homogeneous freezing threshold are often difficult to realize. Several studies describe in

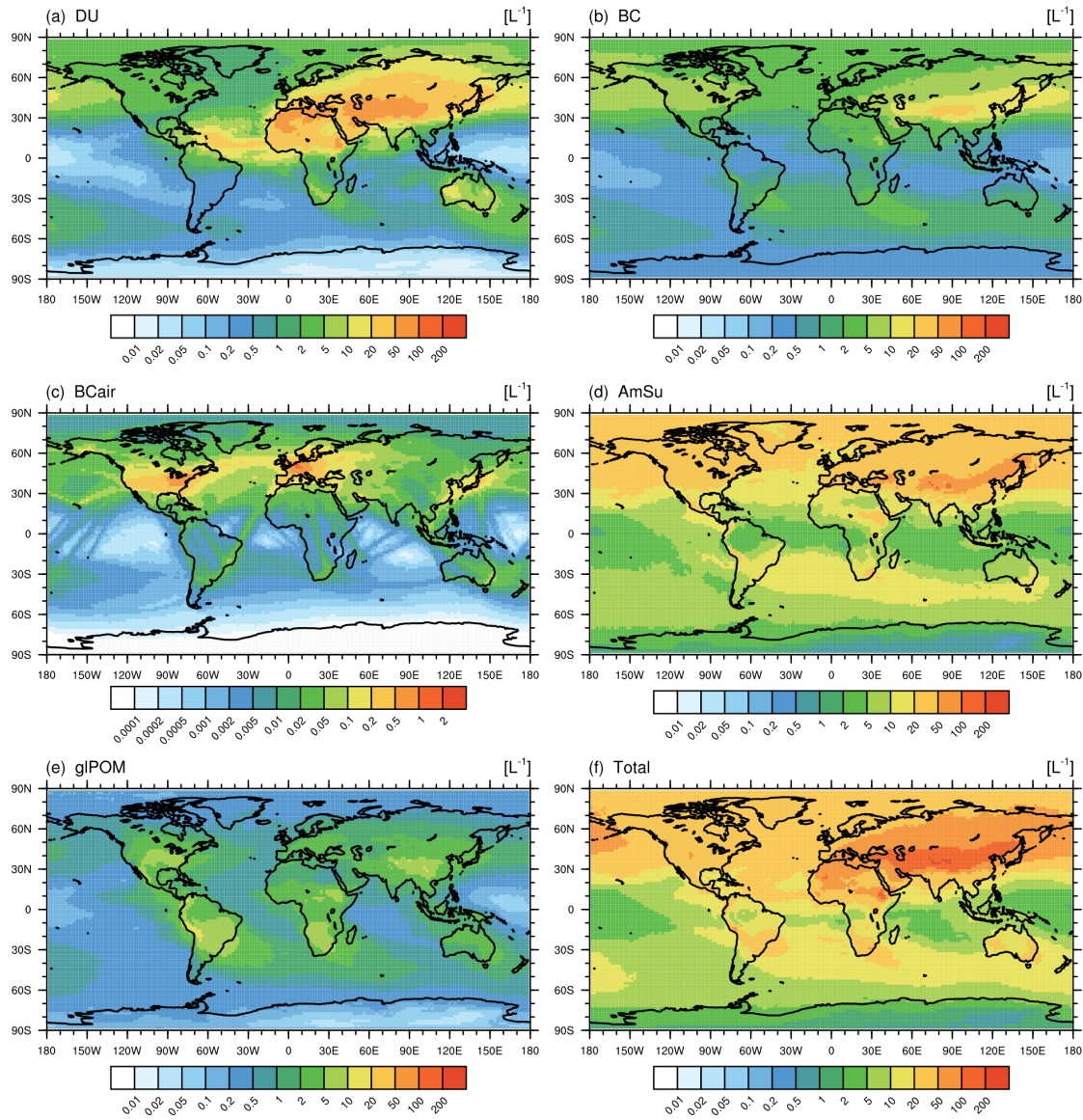

**Figure 5.** Global distribution of simulated number concentrations of different INPs (in units of $L^{-1}$) inside cirrus clouds (selecting only grid boxes with cirrus occurrence) considering the multi-year average over the simulated time period (2010–2019) and over all vertical levels. Shown are (a) mineral dust (DU), (b) black carbon (BC), (c) black carbon from aviation (BCair), (d) ammonium sulfate (AmSu), (e) glassy organics (glPOM), and (f) total INP concentrations. Cirrus conditions are selected according to thresholds for simulated ambient temperature ($T < 238$ K) and ice water content (IWC $> 0.5$ mg kg$^{-1}$) in every grid box using the 11-h output frequency. Number concentrations of potential INPs (see calculations in Sect. 2.1) are weighted with ice-active fractions, $f_{\mathrm{act}}$ at ice supersaturations of $S_i = 1.4$, from laboratory measurements (see Table 1). Note the different scale in panel (c).





situ observations of number concentrations of atmospheric INPs, mostly collected at low altitudes and analysed at temperatures
above 238 K, ranging from concentrations below $0.1\,\mathrm{L}^{-1}$ to several $100\,\mathrm{L}^{-1}$ (e.g., Rogers et al., 1998, 2001a; DeMott et al.,
2010; Schrod et al., 2017), in accordance with INP concentrations simulated here (Fig. 5).

In addition to the global distributions in Fig. 5, a more condensed analysis is presented in Fig. 6a. INP number concentrations
for five different latitude regions are shown as a frequency distribution plot. Frequencies are drawn as shaded colours for every

INP-type and latitude region. Number concentrations of potential INPs are again multiplied with $f_{\mathrm{act}}$ at $S_i = 1.4$, and cirrus
conditions are selected according to temperature and ice water content.

All INP-types except glPOM show a marked difference between Northern and Southern Hemisphere with the highest con-
centrations at the northern latitudes. BC, BCair and AmSu are mostly concentrated in the region of $30°–60°\,\mathrm{N}$ as anthropogenic
influences and emissions play a key role for these INP-types. Aircraft BC has notable concentrations almost only in this region

with concentrations in the range of $10^{-3}$ to $1\,\mathrm{L}^{-1}$ showing the highest occurrence frequencies. In other regions BCair concen-
trations are generally much lower than those of other INP-types. Non-aircraft BC INPs frequently reach concentrations of up
to $10\,\mathrm{L}^{-1}$ in those regions. AmSu INPs can show high concentrations of up to $100\,\mathrm{L}^{-1}$, exceeding in most cases other INP
concentrations. However, in many other cases AmSu concentrations are very low or zero (bottom bins occurring with probabil-
ities of up to 49 %). These are mainly related to ambient conditions where ammonium sulfate does not occur in its crystalline

state. Mineral dust INPs occur most frequently in the latitude region around the equator and $30°–60°\,\mathrm{N}$, where dominant dust
emission regions are situated (e.g. the Sahara, Arabian and Asian deserts). Glassy organic INPs are more evenly distributed
over all latitudes, with concentrations mostly below $5\,\mathrm{L}^{-1}$, lower compared to most other INP-types. Similar to AmSu, ambi-
ent conditions often do not favour the glassy state, which leads to large occurrence frequencies in the lowest concentration bin
(up to 77 %). Additionally, we analyse the INP number concentrations for the three different representations of glass transition

temperatures $T_g(\mathrm{RH})$, considering the different SOA proxies citric acid, glucose, and sucrose, respectively (see Fig. S6 in the
Supplement). Using different $T_g$ representations results in an additional uncertainty in glPOM INP number concentrations of
up to one order of magnitude. Differences are largest between the SOA proxies citric acid and glucose but only slight between
glucose and sucrose.

Figures 5 and 6a can help to identify regions on the globe where different INP-types are likely to compete with each other

for the available supersaturated water vapour during the freezing process. On the Southern Hemisphere a competition between
ammonium sulfate and glassy organics is possible, as these INPs are highly concentrated in southern regions while other INP-
types show low concentrations. The Northern Atlantic is a possible competition region between AmSu, BC, and BCair INPs.
Over most regions in Central Asia several different INP-types (DU, BC, AmSu) are present in relatively similar concentrations
leading to possible competition mechanisms between these INPs.

By coupling the different INP-types to the microphysical cirrus cloud scheme, their ice nucleation potential at cirrus con-
ditions can be analysed, including possible competition mechanisms between different INPs. Similar to INP concentrations,
we show the number concentrations of pristine ice crystals formed via heterogeneous freezing induced by INPs and the cor-
responding ice water content (i.e. cloud ice mass per unit mass of air) in Fig. 6b, c for each heterogeneous freezing mode.
The analysis of these quantities allows for an evaluation of the relative importance of the different INP-types in heterogeneous

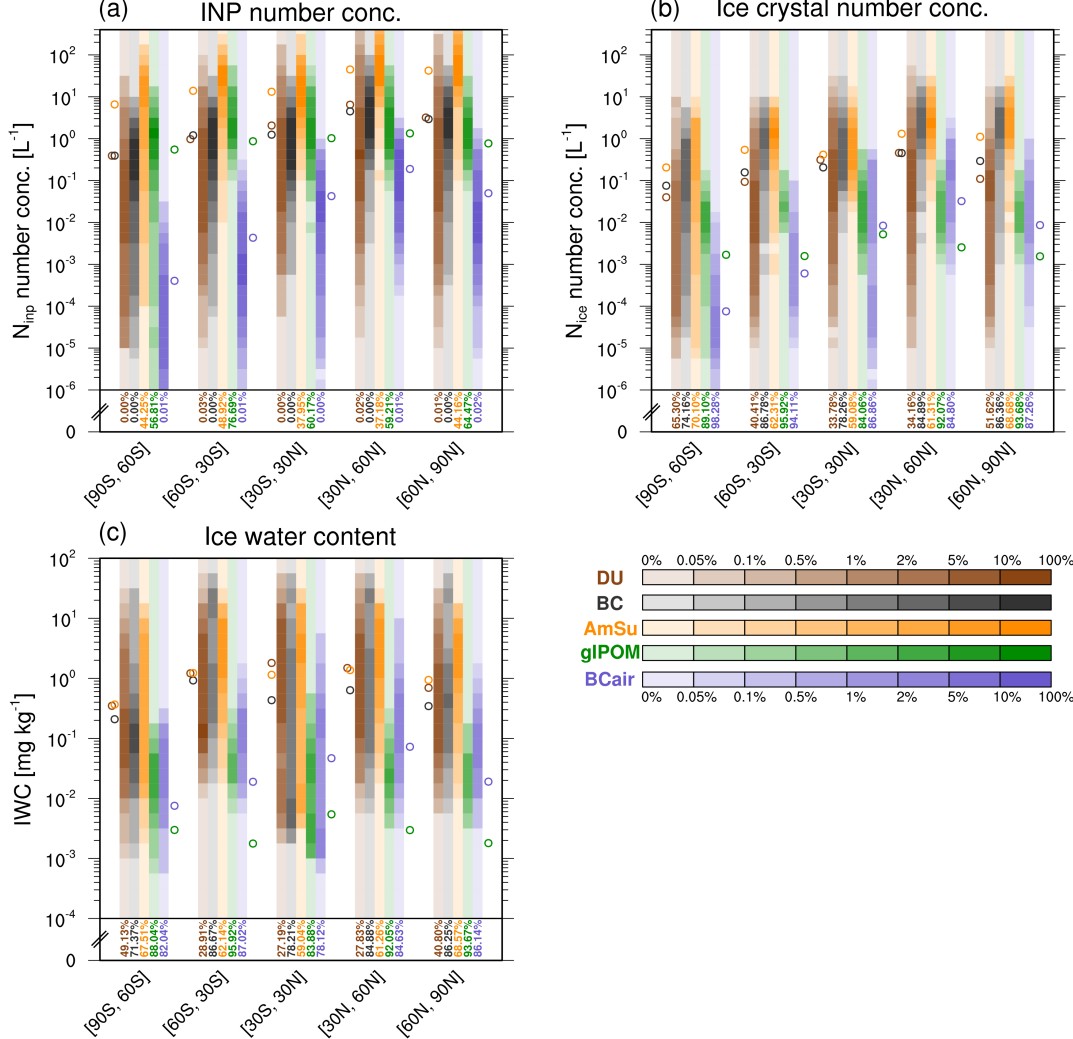

**Figure 6.** Frequency distributions of (a) modelled INP number concentrations (in units of $L^{-1}$), (b) number concentrations of pristine ice crystals ($N_{ice}$; in units of $L^{-1}$), and (c) ice water content (IWC; in units of $mg\,kg^{-1}$) inside cirrus clouds per freezing mode for five different latitude regions, calculated from 11-hour model output for the simulated period 2010–2019. Shaded colours represent the frequency of occurrence within specific bins for the respective variable considering four logarithmic bins per order of magnitude. The different colors refer to DU (brown), BC (black), AmSu (orange), glPOM (green), BCair (purple). Frequencies in the bottom bin, including all values down to zero, are depicted as coloured text. Cirrus conditions are selected according to temperature ($T < 238\,K$) and ice water content (IWC $> 0.5\,mg\,kg^{-1}$) in every model grid box and time step. Additionally, only grid boxes with no contribution of ice-formation on preexisting ice crystals are selected ($N_{preex} < 10^{-23}\,L^{-1}$). Number concentrations of potential INPs are weighted with ice-active fractions from laboratory measurements ($f_{act}$ at $S_i = 1.4$, see Table 1). Open circles to the left and right of every latitude region denote mean values for the different INP-species.





ice formation. Concentrations of newly formed ice crystals from the different freezing modes (Fig. 6b) are generally lower than INP number concentrations (on average about 1–2 orders of magnitude). Additionally, very small values (bottom bins in Fig. 6b) occur much more frequently. This clearly shows that only a fraction of the number of INPs actually nucleates and forms ice crystals. Importantly, the bottom bins (i.e. very small values) for ice crystals from mineral dust freezing show often much lower frequencies compared to the other freezing modes. This indicates a pronounced competition mechanism between the dif-

ferent INPs. The most efficient INPs, i.e. mineral dust and also ammonium sulfate, rapidly deplete the available supersaturated water vapour and inhibit freezing induced by less ice-active INP-types.

In addition to $N_{\mathrm{ice}}$ from heterogeneous freezing modes, the number concentrations of homogeneously formed ice crystals are shown in Fig. 7. In all latitude regions, these are much larger than concentrations of heterogeneously formed ice crystals (up to 3–4 orders of magnitude). Occurrence frequencies in the lowest concentration bin are often very large (78-97 %), indi-

cating that heterogeneous freezing often prevents homogeneous ice nucleation. However, heterogeneous INPs do not suppress homogeneous freezing completely and when homogeneous freezing takes place, it produces very large concentrations of ice crystals. This process was also described in detail in e.g. Kärcher and Lohmann (2002) and Kärcher et al. (2006). The extent of this attenuation effect and resulting cirrus and climate modifications have been shown recently for the case of the aviation soot–cirrus effect (Righi et al., 2021).

Figure 6c shows the ice water content (i.e. cloud ice mass per unit mass of air) for each heterogeneous freezing mode. The IWC of pristine ice crystals indicates how much ice mass accumulates on INPs during the freezing process, regarding the nucleation and growth of these crystals during the first model time step of the existence of the respective cirrus cloud. INPs with a high freezing-efficiency initiate nucleation already at relatively low ice-supersaturations, resulting in ice crystals growing to larger sizes during the freezing process. Here, the highly efficient INPs, i.e. mineral dust and ammonium sulfate,

often dominate the IWC in several regions with average values up to $2 \mathrm{~mg\,kg^{-1}}$, and often show lower frequencies in the lowest concentration bin compared to the other INP-types. This indicates the large importance of these two INP-types. It further reveals that crystalline ammonium sulfate, which has been neglected in most of the previous global modelling efforts on aerosol-induced cirrus formation, needs to be taken into account in future studies.

To evaluate the quality of the simulated INP numbers, they can be compared with in situ observations that, however, mostly

focused on the mixed-phase temperature regime. Rogers et al. (1998) described aircraft measurements performed over North America at high altitudes around $10 \mathrm{~km}$ but analysed samples at temperatures mostly above $243 \mathrm{~K}$ in a flow diffusion chamber. They reported INP concentrations between $\sim 0.1$ and $500 \mathrm{~L^{-1}}$, with larger concentrations at lower temperatures and higher supersaturations. This agrees well with our model results obtained for that region (Fig. 5f). Additionally, Rogers et al. (2001a) analysed aircraft measurements conducted in the Arctic for temperatures between $263 \mathrm{~K}$ and $243 \mathrm{~K}$ and humidities ranging

from ice saturation to water supersaturation. They observed an average INP-concentration of $16 \mathrm{~L^{-1}}$ with rare very high concentration values (hundreds per litre). This is in good accordance with the simulated INP concentrations in the region of $60°$–$90°$ N of around $20 \mathrm{~L^{-1}}$. DeMott et al. (2010) presented a parametrization of INP concentrations using a combination of data from several different aircraft campaigns. Measurements were mostly performed for temperatures above $239 \mathrm{~K}$ where INP concentrations ranging from $0.1$–$100 \mathrm{~L^{-1}}$ were observed, which corresponds to the range of simulated values shown in Fig. 6a.

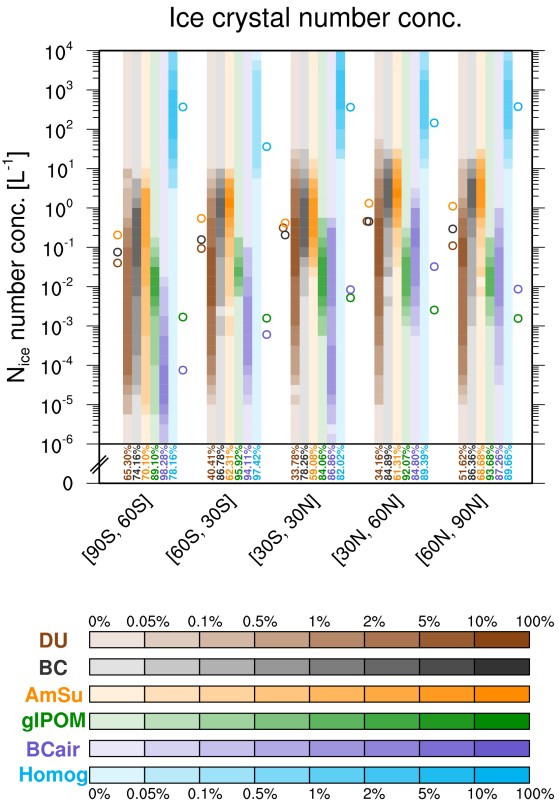

**Figure 7.** As in Fig. 6b, but additionally showing the frequency distribution of ice crystals formed by homogeneous freezing (light blue).

Jensen et al. (2010, 2013) focused on cirrus clouds in the tropical tropopause layer, and predicted that measured ice crystal numbers and cloud optical properties are in accordance with INP concentrations below $20\ \mathrm{L}^{-1}$. This is in agreement with our model results in the equatorial region presented here (Figs. 5, 6a). Additionally, Schrod et al. (2017) presented measurements performed with unmanned aircraft systems over the Eastern Mediterranean at altitudes below $2.5\ \mathrm{km}$ and reported INP peak concentrations of above $100\ \mathrm{L}^{-1}$ at $243\ \mathrm{K}$, also confirming the model results presented in Fig. 5f.

In addition to observations, other global model studies have been performed to analyse the distribution of ice nucleating particles on the globe. Barahona et al. (2010) carried out simulations with the NASA Global Modeling Initiative chemical and transport model (Rotman et al., 2001; Liu et al., 2007), coupled to an analytical ice microphysics parametrization, considering DU, BC, and glassy organics as heterogeneous freezing nuclei. Their results show INP concentrations mostly around $5\ \mathrm{L}^{-1}$ in the equatorial region, which is in accordance with the results presented here (Fig. 6a). Hendricks et al. (2011) performed

a similar study to analyse the effect of INPs on cirrus clouds employing the ECHAM4 general circulation model (Roeckner et al., 1996), considering heterogeneous freezing on DU and BC in addition to homogeneous freezing. They report modelled ice crystal number concentrations formed via heterogeneous freezing between 0.01 and $0.1\ \mathrm{L}^{-1}$ in agreement with the range of $N_{\mathrm{ice}}$ values described in this section (Fig. 6b). However, $N_{\mathrm{ice}}$ from BC freezing in Hendricks et al. (2011) is about 10 times





lower compared to the dust freezing mode. This difference between DU and BC is not visible in the results presented here.
Disparities with respect to Hendricks et al. (2011) could be due to differences in the representation of the atmospheric aerosol, the parametrization of the freezing processes, and different model dynamics due to the use of nudging.

## 6 Conclusions

In this paper we present a global climatology of ice nucleating particles by analysing global model simulations performed with EMAC-MADE3. This climatology comprises the main currently discussed INP-types at cirrus conditions. Despite mineral dust
and black carbon particles, which have been generally considered as INPs in many previous model studies, we additionally include crystalline ammonium sulfate and glassy organic INPs. These two novel INP-species have only rarely been considered in global models, despite several laboratory studies suggesting their pronounced ice-nucleating potential. Hence, their climate impact on a global scale is still highly uncertain.

We present atmospheric distributions of mineral dust and black carbon, including black carbon from aviation emissions,
as also analysed and evaluated in previous studies with EMAC-MADE3 (Beer et al., 2020; Righi et al., 2021). In addition, we describe the implementation of phase transitions of ammonium sulfate and organic particles in detail. The formulation of the phase state is very important for these particles, as only the crystalline phase of ammonium sulfate and the glassy phase of organic particles induce heterogeneous ice-nucleation. We present the atmospheric distributions of crystalline ammonium sulfate and glassy organics and compare their respective atmospheric concentrations with results from previous model studies,
showing overall good agreement with the results presented here.

We calculate the concentrations of potential ice nucleating particles from the simulated aerosol concentrations according to the scheme described in Righi et al. (2020), adapted to include the additional INP-types, i.e. crystalline ammonium sulfate and glassy organics. We present and analyse the resulting multi-annual mean climatology of INPs at cirrus formation conditions. Simulated INP number concentrations, in the range of about 1 to $100 \, \mathrm{L}^{-1}$, agree well with in situ observations (e.g., Rogers
et al., 1998, 2001a; DeMott et al., 2010; Schrod et al., 2017) and other global model studies (e.g., Barahona et al., 2010; Hendricks et al., 2011).

High dust INP concentrations (about $100 \, \mathrm{L}^{-1}$) are simulated over dust-dominated regions (e.g. the Sahara, Arabian, and Asian deserts). Black carbon and ammonium sulfate INPs show a distinct north-south gradient with largest concentrations on the Northern Hemisphere (up to $50 \, \mathrm{L}^{-1}$ and $100 \, \mathrm{L}^{-1}$, respectively), probably dominated by anthropogenic influences. Glassy
organic INPs are concentrated in regions with strong biogenic activity mainly in the tropics and on the Southern Hemisphere, but have often much lower concentrations compared to most other INP-types, i.e. below $10 \, \mathrm{L}^{-1}$. However, these mean values for glassy organics and for ammonium sulfate include also cases where these INPs are not glassy or crystalline, respectively, and larger concentrations are possible at specific times and locations. Aviation soot INPs show highest concentrations along typical aircraft flight routes, mainly between Europe and North America (about $1 \, \mathrm{L}^{-1}$).
By coupling the different INP-types to the microphysical cirrus cloud scheme, their ice nucleation potential at cirrus conditions is analysed, including possible competition mechanisms between different INPs and considering regional, latitude-



specific differences. Concentrations of freshly nucleated pristine ice crystals from heterogeneous freezing are typically one to two orders of magnitude lower than respective INP number concentrations. In many cases, only a fraction of the available INPs can be activated due to low cooling rates. Owing to the comparably small number concentrations of INPs, the mean number

concentrations of INP-induced ice crystals are also much lower than those of homogeneously formed crystals (up to 3–4 orders of magnitude). The most abundant INP-types, e.g. soot, dust, and ammonium sulfate often show the largest impact on the ice crystal number concentration of freshly nucleated crystals. The different INP-species compete with each other for the available supersaturated water vapour. The most efficient INPs, i.e. mineral dust and also ammonium sulfate, in many cases inhibit freezing induced by less ice-active INP-types, leading to frequently occurring very low pristine ice crystal concentrations for

(aviation) soot and glassy organic INPs. The highly efficient dust and ammonium sulfate INPs also generate the largest ice mass and show a marked effect on the ice water content.

To conclude, the climatology of ice nucleating particles at cirrus formation presented in this study demonstrates the importance of including additional ice nucleating particle types together with mineral dust and soot particles in global models. Especially crystalline ammonium sulfate shows a large potential for cirrus cloud modifications due to its large INP concentra-

tions, while glassy organic particles probably have only minor influences, as their concentrations are mostly small.

The new insights achieved in the present study could be further deepened by additional model developments and analyses. Recent laboratory studies identified yet another type of ice nucleating particles at cirrus conditions, i.e. marine aerosol (e.g. Wilbourn et al., 2020; Wagner et al., 2021). Potential influences of these marine particles on cirrus clouds could be evaluated by including them as INPs in the cirrus ice nucleation scheme. Additionally, the analysis of the simulated global distribution

of different INPs could be further refined by applying machine learning clustering algorithms (e.g. k-means clustering; Hartigan and Wong, 1979) to identify regions dominated by specific INP-types or by possible competition mechanisms between different INPs. This technique has recently been demonstrated for the analysis of global aerosol simulations by Li et al. (2022). Following the results presented in this study, cirrus cloud and resulting climate modifications induced by the ice nucleating particles described here could be analysed in order to provide further insight into these INP-induced climate effects.

*Code and data availability.* MESSy is continuously developed and applied by a consortium of institutions. The usage of MESSy, including MADE3, and access to the source code is licensed to all affiliates of institutions which are members of the MESSy Consortium. Institutions can become members of the MESSy Consortium by signing the MESSy Memorandum of Understanding. More information can be found on the MESSy Consortium Website (http://www.messy-interface.org, last access: 26 July 2022). The model configuration discussed in this paper has been developed based on version 2.55 and will be part of the next EMAC release (version 2.56). The exact code

version used to produce the result of this paper is archived at the German Climate Computing Center (DKRZ) and can be made available to members of the MESSy community upon request. The model setup and the simulation data analysed in this work are available at http://doi.org/10.5281/zenodo.6834299 (Beer, 2022).





## Appendix A: Ammonium sulfate phase transition

A schematic overview of the representation of ammonium sulfate in EMAC-MADE3 and its phase transition formulation
is presented in Fig. A1. Calculations concerning ammonium sulfate are performed by the subroutines `update_amsu` and
`amsu_phasetrans`, which are part of the MADE3 submodel and require input from the MESSy submodels PTRAC,
MADE3, and CLOUD. A detailed description of these calculations is presented below.

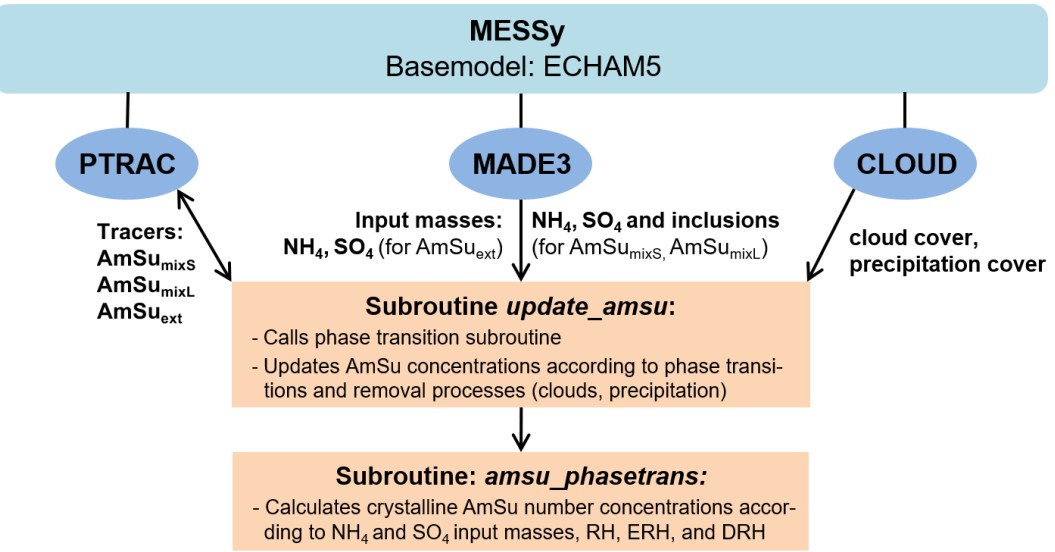

**Figure A1.** Schematic overview of the representation of AmSu tracers in EMAC-MADE3. The subroutines `update_amsu` and
`amsu_phasetrans` are part of the MADE3 submodel and interact with the MESSy submodels PTRAC and CLOUD. The corresponding
AmSu tracers for the coarse mode, i.e. $\mathrm{AmSu_{mix,coa}}$ and $\mathrm{AmSu_{ext,coa}}$, are not shown for the sake of simplicity.

Crystalline ammonium sulfate numbers are calculated according to $\mathrm{NH_4}$ and $\mathrm{SO_4}$ input masses from MADE3, the relative
humidity, ERH and DRH. In order to exclude grid boxes with no or only little $\mathrm{NH_4}$ and $\mathrm{SO_4}$ aerosol mass contribution (for the
case of $\mathrm{AmSu_{mixS}}$, $\mathrm{AmSu_{mixL}}$ tracers), only those grid boxes are taken into account, where the non-dust fraction is dominated
by $\mathrm{NH_4}$ and $\mathrm{SO_4}$, i.e. where $(m_{\mathrm{NH_4}} + m_{\mathrm{SO_4}})/(m_{\mathrm{tot}} - m_{\mathrm{DU}}) \geq 0.5$. This fraction is calculated from the corresponding input
masses taken from MADE3. Dust plays a special role as it facilitates crystallization (e.g. Ushijima et al., 2018). Therefore, if
the non-$\mathrm{NH_4}$-$\mathrm{SO_4}$-mass is dominated by dust, i.e. $m_{\mathrm{DU}}/(m_{\mathrm{tot}} - m_{\mathrm{NH_4}} - m_{\mathrm{SO_4}}) \geq 0.5$, this effect is taken into account by an
increase in ERH, as described below.
Additionally, the ammonium input mass is limited, if there is too much ammonium compared to the available sulfate, i.e.
$\mathrm{NH_4}$ is scaled by $1/\mathrm{ASR}$ (for $\mathrm{ASR} > 1$), where ASR is the ammonium-to-sulfate molar ratio defined as

$$\mathrm{ASR} = \frac{[\mathrm{NH_4}]}{2\,[\mathrm{SO_4}]} \; . \tag{A1}$$





For stoichiometric reasons, this reduction represents the maximum amount of ammonium which can be transformed to the solid phase. For the case of $\mathrm{AmSu_{mixS}}$ and $\mathrm{AmSu_{mixL}}$, the mass of non-$\mathrm{NH_4}$-$\mathrm{SO_4}$ inclusions is calculated as

$$m_{\mathrm{incl}} = m_{\mathrm{tot}} - m_{\mathrm{SO_4}} - m_{\mathrm{NH_4}} \, . \tag{A2}$$

In addition to $\mathrm{NH_4}$, the inclusion mass has to be scaled (if $\mathrm{ASR} > 1$) by

$$\frac{m_{\mathrm{SO_4}} + m_{\mathrm{NH_4}}/\mathrm{ASR}}{m_{\mathrm{SO_4}} + m_{\mathrm{NH_4}}} \, , \tag{A3}$$

which represents the reduction of the $(\mathrm{SO_4} + \mathrm{NH_4})$-mass. ASR is subsequently limited to $0 \leq \mathrm{ASR} \leq 1$.

The deliquescence relative humidity is set to $\mathrm{DRH} = 80\,\%$ (Martin, 2000). Temperature dependency of DRH can be ne-
glected, as it has only a minor influence (Onasch et al., 1999; Tang and Munkelwitz, 1993). The efflorescence relative humidity is calculated as a function of ASR according to Martin et al. (2003) and Wang et al. (2008).

$$\mathrm{ERH(ASR)} = \begin{cases} -71925 + 1690\,\mathrm{ASR} - 139\,\mathrm{ASR}^2 + \frac{1770760}{25.0 + 0.5(\mathrm{ASR} - 0.7)} \, , & \text{if } 0.5 \leq \mathrm{ASR} \leq 1.0 \\ 0 \, , & \text{if } \mathrm{ASR} < 0.5 \end{cases} \tag{A4}$$

By setting ERH to zero if $\mathrm{ASR} < 0.5$, cases where there is too much $\mathrm{SO_4}$ for the available $\mathrm{NH_4}$ are excluded. This implies that no crystallization is possible for $\mathrm{ASR} < 0.5$, as reported in Martin et al. (2003) and Wang et al. (2008). The temperature
dependency of ERH is small and can be neglected (Onasch et al., 1999).

Equation (A4) provides a minimum ERH for the case of pure ammonium sulfate particles (without inclusions). As stated in various studies (e.g., Martin, 2000; Martin et al., 2001; Han et al., 2002; Martin et al., 2003; Pant et al., 2006; Ushijima et al., 2018), inclusions of e.g. mineral dust can increase the ERH significantly, leading to more favourable conditions for the crystalline phase. The effect of mineral dust is enhanced for larger inclusion diameters. According to Martin et al. (2003) this
inclusion effect (heterogeneous nucleation) can be added to Eq. (A4) by:

$$\mathrm{ERH_{het}(ASR}, D) = \mathrm{ERH_{hom}(ASR)} + f(D) \, , \tag{A5}$$

with $f(D)$ being a function of the inclusion diameter $D$. Here, the effect of mineral dust inclusions is implemented in the following way. For $\mathrm{AmSu_{mixL}}$, ERH is increased by $10\,\%$, if dust inclusions are present in a significant amount (dust fraction with respect to non-$\mathrm{SO_4}$-$\mathrm{NH_4}$ components $> 0.5$). This is in accordance with measurements from Ushijima et al. (2018) and
Pant et al. (2006), where ERH is increased by approximately $10\,\%$ compared to homogeneous efflorescence for dust inclusions with diameters of about $400\,\mathrm{nm}$ (corresponding to typical MADE3 accumulation mode dust particle sizes). For simplicity, this ERH increase is assumed to be the same for $\mathrm{AmSu_{mixS}}$.

The actual ammonium sulfate phase transition is calculated in the following way. If the local $\mathrm{RH} < \mathrm{ERH}$, the total $\mathrm{NH_4}$ and $\mathrm{SO_4}$ mass (with possible inclusions of other components for the case of $\mathrm{AmSu_{mixS}}$ and $\mathrm{AmSu_{mixL}}$) is used to update
the AmSu mass. Subsequently, this is converted to a number concentration by using the conversion function for lognormal distributions (e.g., Seinfeld and Pandis, 2016)

$$\mathrm{C}(D, \sigma_{\mathrm{g}}) = \frac{6}{\pi} \frac{1}{D^3 \exp(4.5 \ln^2 \sigma_{\mathrm{g}}) \, \rho} \, , \tag{A6}$$





where $\rho$ is the component-mass-weighted mean particle density according to the mean particle density calculated in MADE3, $\sigma_g$ and $D$ are the geometric standard deviation and diameter input values specified for the AmSu tracers. In case of deliques-

cence, AmSu is set to zero if RH > DRH.

As AmSu tracers are removed in clouds (see the following section), the relative humidity in the cloud-free area of the grid box ($RH_0$) is used for RH. $RH_0$ can be calculated from the grid box mean RH ($RH_m$) according to the following equations.

$$Q_m = (1 - C_{cl})\, Q_0 + C_{cl}\, Q_s \Rightarrow Q_0 = (Q_m - C_{cl}\, Q_s)/(1 - C_{cl}) \tag{A7a}$$

$$RH_0 = \frac{Q_0}{Q_s} = (RH_m - C_{cl})/(1 - C_{cl})\,, \tag{A7b}$$

with the grid box mean specific humidity $Q_m$, the cloud-free specific humidity $Q_0$, the in-cloud specific humidity $Q_s$ assumed to be equal to the saturation specific humidity, and the cloud covered fraction of the grid box $C_{cl}$.

**Ammonium sulfate removal by clouds and precipitation**

The use of the scavenging submodel SCAV is deactivated for the AmSu tracers, since a very detailed representation of different scavenging processes is not considered necessary in this context, due to a high probability of deliquescence of crystalline am-

monium sulfate in the presence of clouds and precipitation. Instead, concentration changes of AmSu tracers due to interactions with clouds and precipitation are dealt with in the following way.

AmSu is set to zero in the presence of clouds or precipitation as this implies a high probability for deliquescence due to the high relative humidity. This applies for liquid clouds, but also for the mixed phase regime where ice originates from freezing of liquid water. In the cirrus regime a different removal process is applied (as described below). Since the presence of clouds and

precipitation is an indicator for enhanced relative humidity within the corresponding grid box and the simulation of clouds and precipitation is highly uncertain in global climate models, it is assumed that deliquescence occurs in the whole grid box as soon as the fraction of the grid box covered by clouds or precipitation exceeds a critical value. Here, we assume a comparatively conservative threshold value of 10 %. This conservative approach is followed (in contrast to scaling ammonium sulfate tracer changes with the cloud and precipitation free area fraction of the grid box) to avoid overestimations of crystalline ammonium

sulfate numbers.

In addition to the treatment of stratiform cloud effects described above, convective clouds are dealt with by setting an additional threshold (0.1 %) for convective cloud cover ($C_{cv}$, diagnostic model output estimated from the updraft strength) and corresponding removal of AmSu. The simulations are not very sensitive to this threshold, i.e. changes in AmSu concentrations of mostly a few percent when increasing the threshold for $C_{cv}$ by a factor of ten. The information about cloud-covered and

precipitation-covered fraction of the grid box is taken from the CLOUD submodel. A random overlap of cloud cover and precipitation cover within a grid box is assumed, i.e. the area covered by clouds or precipitation ($C_{cl,pr}$) is given by:

$$C_{cl,pr} = C_{cl} + C_{pr} - C_{cl} \cdot C_{pr}\,, \tag{A8}$$

with the area of the grid box covered with clouds ($C_{cl}$) and precipitation ($C_{pr}$), respectively.





A caveat of this procedure is that the cloud cover does not differentiate between liquid and ice clouds. For ice clouds there

may possibly be no deliquescence at all. Hence, an alternative concept for handling ammonium sulfate in the cirrus regime has

been implemented as described below. The same issue applies to precipitation cover. Deliquescence will happen for the case

of rain, but possibly not for snow. In addition, evaporation of snow could lead to a release of AmSu. However, this process

is assumed to be not very relevant for ice nucleation processes as, in many cases, those particles are released well below the

cirrus level.

In cirrus clouds the relative humidity with respect to ice is potentially lower than the deliquescence humidity of ammonium

sulfate, so that AmSu crystals can persist. Therefore, AmSu should be removed in the cirrus regime ($T < T_{\mathrm{hom}} = 238\,\mathrm{K}$)

for non-deliquescence conditions only if snow formation takes place. In addition, cloud ice could possibly release crystalline

ammonium sulfate after evaporation. Here, AmSu removal in cirrus clouds is implemented in the following way: The AmSu

fraction that is removed by snow is calculated from the snow formation rate ($R_{\mathrm{snow}}$) and the cloud ice water content (IWC)

taken from the CLOUD submodel, cloudcover ($C_{\mathrm{cl}}$), and ice active fraction of AmSu ($f_{\mathrm{act}}$):

$$f_{\mathrm{snow}} = \frac{R_{\mathrm{snow}}}{\mathrm{IWC}} \cdot C_{\mathrm{cl}} \cdot f_{\mathrm{act}} \tag{A9}$$

$f_{\mathrm{act}}$ is set to 0.01, according to typical literature values (e.g., Ladino et al., 2014). Subsequently, AmSu concentrations are

scaled with $(1 - f_{\mathrm{snow}})$. In this approach, the effect of snow impaction scavenging of AmSu particles is neglected.

**Appendix B: Calculation of numbers of ice nucleating aerosol particles**

In Righi et al. (2020) the calculations for different freezing mechanisms in the mixed-phase and cirrus regime considering

mineral dust and soot INPs are described in detail. Here, the focus is on the addition of glPOM and AmSu freezing in the cirrus

regime. The existing formulation for tagged BC particles is used and adapted for BC emissions from aviation. The number of

particles available for immersion freezing in mixed-phase clouds is estimated as a fraction of the number of aerosol particles

activated to form cloud droplets ($N_{\mathrm{act}}$), as described in Righi et al. (2020). The number concentrations of potential INPs

available for deposition nucleation and immersion freezing in cirrus clouds are indicated by $N^{\mathrm{dep(c)}}$ and $N^{\mathrm{imm(c)}}$, respectively.

For AmSu and glPOM the freezing mode (deposition or immersion freezing) is uncertain and the number concentrations are

defined more generally as $N^{(\mathrm{c})}$. The number concentrations of actual INPs are then calculated according to their freezing

spectra depending on $S_c$ and $f_{\mathrm{act}}$ applying the Kärcher et al. (2006) parametrization.

Following the same notation as in Kaiser et al. (2019), MADE3 Aitken, accumulation, and coarse modes are indicated with

the indices $k$, $a$, and $c$, respectively. Mixing states are depicted by $s$, $i$, and $m$ for soluble, insoluble, and mixed, respectively.

All the calculated number concentrations undergo consistency checks in the code, to make sure that the estimated number

concentrations in each mode are positive and do not exceed the total number concentration in the mode itself.

The INP properties of the mixed and insoluble Aitken modes, where no dust is present, are controlled by BC particles. The

freezing efficiency of AmSu and glPOM in the Aitken modes is assumed to be low, due to the small particle size, and these

INPs are neglected for the Aitken modes. In the cirrus regime, first the numbers of BCair INPs are calculated from their masses





in the two different Aitken modes using the mass-to-number conversion factors from Eq. (A6) and assuming size distribution parameters typical for aircraft soot, i.e. $D = 0.025\,\mu\text{m}, \sigma_g = 1.55$ for the Aitken mode and $\rho = 1500\;\text{kg/m}^3$ (Petzold et al., 1999). Since no dust is present in this mode, the remaining number of particles in the mode is then assigned to BC (from non-aircraft sources).

$$N_{\text{BCair,k}}^{\text{imm(c)}} = M_{\text{BCair,km}} C_{\text{BCair,k}} \tag{B1a}$$

$$N_{\text{BCair,k}}^{\text{dep(c)}} = M_{\text{BCair,ki}} C_{\text{BCair,k}} \tag{B1b}$$

$$N_{\text{BC,k}}^{\text{imm(c)}} = \max(0, N_{\text{km}} - N_{\text{BCair,k}}^{\text{imm(c)}}) \tag{B1c}$$

$$N_{\text{BC,k}}^{\text{dep(c)}} = \max(0, N_{\text{ki}} - N_{\text{BCair,k}}^{\text{dep(c)}}) \tag{B1d}$$

The formulations and representations assumed for potential INPs in the accumulation modes are depicted schematically in Fig. B1. The corresponding calculations are summarized in Table B1 and Table B2 and are described in detail below.

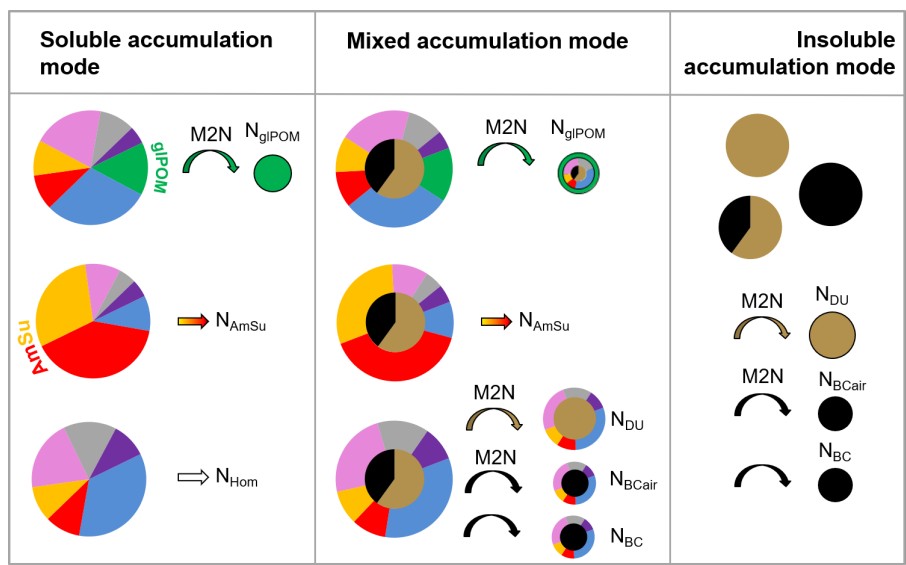

**Figure B1.** Schematic representation of assumed potential INPs concerning the calculation of number concentrations available for freezing processes in the accumulation modes. In the Aitken and coarse modes other sets of possible INP-types are participating, e.g. no AmSu and glPOM freezing in the Aitken modes, and no BCair freezing in the coarse mode is assumed. M2N represents the conversion of aerosol mass to number concentrations using Eq. (A6). Ammonium sulfate is represented in red and yellow, BC and DU in black and brown, respectively. In addition to glPOM (green), other POM may also be present but is omitted here, for the sake of simplicity. Other aerosol species simulated by MADE3 are nitrate (purple), sodium (gray), chloride (pink), and aerosol water (blue).


In the soluble accumulation mode, no DU or BC are present. The number concentration of glPOM potential INPs is calculated from the glPOM mass according to Eq. (A6), assuming $D = 0.2\,\mu\text{m}, \sigma_g = 2.0$ (this represents typical accumulation mode





sizes and is chosen to be comparable to the assumptions for AmSu) and $\rho = 1000 \ \mathrm{kg/m^3}$ (as for the MADE3 POM tracer).

$$N_{\mathrm{glPOM,as}}^{(c)} = M_{\mathrm{glPOM,as}} \, C_{\mathrm{glPOM,as}} \tag{B2}$$

For ammonium sulfate freezing, the number concentration of the AmSu tracers in the mode is used. The contribution of the $\mathrm{AmSu_{mixS}}$ and $\mathrm{AmSu_{mixL}}$ tracers for this mode is calculated according to the MADE3 $\mathrm{NH_4}$ and $\mathrm{SO_4}$ mass concentrations.

$$N_{\mathrm{AmSu_{mix}}}^{\mathrm{as}} = f_{\mathrm{as}} \cdot (N_{\mathrm{AmSu_{mixS}}} + N_{\mathrm{AmSu_{mixL}}}) \ , \tag{B3}$$

where $f_{\mathrm{as}}$ is the fractional contribution of the soluble accumulation mode to the total accumulation mode $\mathrm{NH_4}$ and $\mathrm{SO_4}$ mass, calculated as

$$f_{\mathrm{as}} = \frac{m_{\mathrm{NH_4}}^{\mathrm{as}} + m_{\mathrm{SO_4}}^{\mathrm{as}}}{m_{\mathrm{NH_4}}^{\mathrm{as}} + m_{\mathrm{SO_4}}^{\mathrm{as}} + m_{\mathrm{NH_4}}^{\mathrm{am}} + m_{\mathrm{SO_4}}^{\mathrm{am}}} \ . \tag{B4}$$

Subsequently, this number is reduced by the number of glPOM INPs in the mode (assuming a maximum neutralization efficiency of the AmSu INPs due to glPOM shells formed by interactions of AmSu with organic aerosols) to obtain the number of potential AmSu INPs in the mode.

$$N_{\mathrm{AmSu,as}}^{(c)} = \max(0, N_{\mathrm{AmSu_{mix}}}^{\mathrm{as}} - N_{\mathrm{glPOM,as}}^{(c)}) \tag{B5}$$

Additionally, the number concentration of soluble aerosols available for homogeneous freezing in the mode ($N_{\mathrm{hom,as}}$) is reduced by AmSu and glPOM numbers. For AmSu this reduction is dependent on the relation between the ice supersaturation for homogeneous freezing ($S_{\mathrm{hom}}$; Koop et al., 2000) and the deliquescence relative humidity. If $S_{\mathrm{hom}} \geq \mathrm{DRH}$ only the ice active fraction of AmSu ($f_{\mathrm{act}}$) is subtracted, as the remaining particles would be liquid and available for homogeneous freezing.

$$N_{\mathrm{hom,as}} = \begin{cases} \max(0, N_{\mathrm{as}} - N_{\mathrm{glPOM,as}}^{(c)} - N_{\mathrm{AmSu_{mix}}}^{\mathrm{as}}) \ , & \text{if } S_{\mathrm{hom}} < \mathrm{DRH} \\ \max(0, N_{\mathrm{as}} - N_{\mathrm{glPOM,as}}^{(c)} - f_{\mathrm{act}} \cdot N_{\mathrm{AmSu_{mix}}}^{\mathrm{as}}) \ , & \text{if } S_{\mathrm{hom}} \geq \mathrm{DRH} \end{cases} \tag{B6}$$

In the mixed accumulation mode all possible INPs can be present, i.e. DU, BC, AmSu, glPOM. First, the numbers of glassy organics are calculated from their mass, assuming a 50 nm thick spherical glPOM shell around a 100 nm core. This size corresponds to the assumption used for AmSu particle sizes, to keep these INP-types comparable. Following a simple geometric calculation, such a shell has an equivalent volume as a sphere with $D = 191$ nm. Together with $\sigma_g = 2.0$ and $\rho = 1000 \ \mathrm{kg/m^3}$
this diameter is used to convert glPOM mass to number concentration using Eq.(A6).

$$N_{\mathrm{glPOM,am}}^{(c)} = M_{\mathrm{glPOM,am}} \, C_{\mathrm{glPOM,am}} \tag{B7}$$

The number concentration of potential AmSu INPs is again calculated as the fraction of AmSu tracers in this mode:

$$N_{\mathrm{AmSu,am}}^{(c)} = (1 - f_{\mathrm{as}}) \cdot (N_{\mathrm{AmSu_{mixS}}} + N_{\mathrm{AmSu_{mixL}}}) \tag{B8}$$



The numbers of potential BCair and DU INPs are calculated similarly, with the assumption of $D = 0.15$ μm, $\sigma_g = 1.65$ and
$\rho = 1500$ kg/m$^3$ for BCair (Petzold et al., 1999) and $D = 0.42$ μm, $\sigma_g = 1.59$ and $\rho = 2500$ kg/m$^3$ for DU following the
AeroCom recommendations (Dentener et al., 2006). These parameters for DU are the same that are also used to calculate
the number of emitted dust particles in the model (see Beer et al., 2020, for details), i.e. the ageing of dust particles due to
coagulation is neglected, which is regarded as a viable simplification due to the comparably low number concentration of dust
particles in the atmosphere and the resulting low coagulation efficiency. Additionally, the numbers of DU and BCair immersion
freezing INPs have to be reduced according to the respective inclusions inside AmSu particles ($N_{\text{incl}}^{\text{DU}}, N_{\text{incl}}^{\text{BCair}}$). Therefore, DU
and BCair numbers are calculated as:

$$N_{\text{DU,am}}^{\text{imm(c)}} = \max(0, M_{\text{DU,am}} C_{\text{DU,a}} - N_{\text{incl}}^{\text{DU}}) \tag{B9a}$$

$$N_{\text{BCair,am}}^{\text{imm(c)}} = \max(0, M_{\text{BCair,am}} C_{\text{BCair,a}} - N_{\text{incl}}^{\text{BCair}}) \ , \tag{B9b}$$

with

$$N_{\text{incl}}^{\text{DU}} = \frac{N_{\text{DU,am}}}{N_{\text{am}}} \cdot N_{\text{AmSu,am}}^{\text{imm(c)}} \tag{B10a}$$

$$N_{\text{incl}}^{\text{BCair}} = \frac{N_{\text{BCair,am}}}{N_{\text{am}}} \cdot N_{\text{AmSu,am}}^{\text{imm(c)}} \ . \tag{B10b}$$

Subsequently, the numbers of potential DU, BCair, and AmSu INPs have to be reduced according to glPOM numbers and the
assumption of glPOM forming a shell around other particles (e.g. Smith et al., 2012, 2013; Schill et al., 2014; Saukko et al.,
2015). Their numbers are reduced by multiplication with the scaling factor

$$f_{\text{glPOM}}^{\text{am}} = (N_{\text{am}} - N_{\text{glPOM,am}}^{(c)})/N_{\text{am}} \ . \tag{B11}$$

The number of BC particles available for immersion freezing is finally calculated as the remaining number in the mode:

$$N_{\text{BC,am}}^{\text{imm(c)}} = \max(0, N_{\text{am}} - N_{\text{DU,am}}^{\text{imm(c)}} - N_{\text{BCair,am}}^{\text{imm(c)}} - N_{\text{glPOM,am}}^{(c)} - N_{\text{AmSu,am}}^{(c)}) \tag{B12}$$

Here, BC particles are assumed to have accumulation mode sizes. Note that BC particles in the mixed accumulation mode
not necessarily show accumulation mode sizes, since this mode may consist of a mixture of particles from the soluble accu-
mulation and insoluble or mixed Aitken mode BC or BCair particles. This should be considered in the representation of the
heterogeneous BC ice nucleation.

In the insoluble accumulation mode only DU, BCair and BC are present. The numbers of DU and BCair deposition freezing
INPs are here again calculated from their masses, and the remaining particles in the mode are ascribed to BC deposition
freezing.

$$N_{\text{DU,a}}^{\text{dep(c)}} = M_{\text{DU,ai}} C_{\text{DU,a}} \tag{B13a}$$

$$N_{\text{BCair,a}}^{\text{dep(c)}} = M_{\text{BCair,ai}} C_{\text{BCair,a}} \tag{B13b}$$

$$N_{\text{BC,a}}^{\text{dep(c)}} = \max(0, N_{\text{ai}} - N_{\text{DU,a}}^{\text{dep(c)}} - N_{\text{BCair,a}}^{\text{dep(c)}}) \tag{B13c}$$



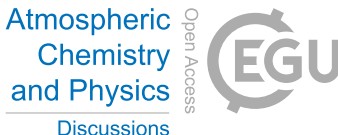

Calculations for the soluble coarse mode, concerning immersion freezing of glPOM and AmSu, and homogeneous freezing of solution droplets, are performed similar to those in the soluble accumulation mode, but with different assumptions for the parameters used for mass-to-number conversion of glPOM particles, i.e. $D = 2.0\,\mu\text{m}, \sigma_g = 2.2$ and $\rho = 1000\,\text{kg/m}^3$.

$$N^{(c)}_{\text{glPOM,cs}} = M_{\text{glPOM,cs}} C_{\text{glPOM,cs}} \tag{B14}$$

For ammonium sulfate freezing the fraction of the $\text{AmSu}_{\text{mix,coa}}$ tracer in the mode is used including the reduction according to glPOM numbers in the mode (assuming a maximum neutralization efficiency of the AmSu INPs due to glPOM shells).

$$N^{\text{cs}}_{\text{AmSu}_{\text{mix,coa}}} = f_{\text{cs}} \cdot N_{\text{AmSu}_{\text{mix,coa}}} \tag{B15a}$$

$$N^{(c)}_{\text{AmSu,cs}} = \max(0, N^{\text{cs}}_{\text{AmSu}_{\text{mix,coa}}} - N^{(c)}_{\text{glPOM,cs}})\ , \tag{B15b}$$

where $f_{\text{cs}}$ is the fractional contribution of the soluble coarse mode to the total accumulation mode $\text{NH}_4$ and $\text{SO}_4$ mass, in analogy to $f_{\text{as}}$ in the accumulation modes (see Eq. B4).

Additionally, the number concentration of soluble aerosols available for homogeneous freezing in the mode ($N_{\text{hom,cs}}$) is reduced by AmSu and glPOM numbers depending on $S_{\text{hom}}$, DRH, and the ice active fraction of AmSu ($f_{\text{act}}$) as for the soluble accumulation mode.

$$N_{\text{hom,cs}} = \begin{cases} \max(0, N_{\text{cs}} - N^{(c)}_{\text{glPOM,cs}} - N^{\text{cs}}_{\text{AmSu}_{\text{mix,coa}}})\ , & \text{if } S_{\text{hom}} < \text{DRH} \\ \max(0, N_{\text{cs}} - N^{(c)}_{\text{glPOM,cs}} - f_{\text{act}} \cdot N^{\text{cs}}_{\text{AmSu}_{\text{mix,coa}}})\ , & \text{if } S_{\text{hom}} \geq \text{DRH} \end{cases} \tag{B16}$$

In the mixed coarse mode DU, BC, glPOM, and AmSu INPs may be present. In this mode it is assumed that there is no contribution of BCair particles to the number concentrations since these particles are too small to cause a mass dominance within the INP material in the coarse mode. A similar calculation as in the mixed accumulation mode is performed. Here, glPOM is assumed to form a 0.5 µm thick spherical shell around a 1 µm core, which has an equivalent volume (or mass) as a sphere with $D = 1.91\,\mu\text{m}, \sigma_g = 2.2$ and $\rho = 1000\,\text{kg/m}^3$. For DU, $D = 1.3\,\mu\text{m}, \sigma_g = 2.0$ and $\rho = 2500\,\text{kg/m}^3$ are used (Dentener et al., 2006). However, as in this mode particles can be composed of dust from both the accumulation and coarse size ranges, whose relative contribution is unknown, two cases are distinguished, in analogy to Righi et al. (2020), according to the relative abundance of mineral dust. Defining the dust number fraction as

$$f_{\text{DU}} = \frac{M_{\text{DU,cm}} C_{\text{DU,c}}}{N_{\text{cm}}}\ , \tag{B17}$$

it is assumed that for large $f_{\text{DU}}$ the mode is dominated by mineral dust. Here, coarse dust particles are assumed to calculate the number fraction, as these particles dominate the dust mass (possible mass contributions of accumulation-mode dust are small, according to Dentener et al. (2006)). For large $f_{\text{DU}}$ it can be expected that other INPs have a relatively small contribution, and all particles in the mode can be regarded as possible DU INPs. Here, $f_{\text{DU}} \geq 0.7$ is assumed as a dominance threshold for DU[2].

---

[2] Righi et al. (2020) performed a sensitivity study and showed that lower threshold values for $f_{\text{DU}}$ (0.5 and 0.6) do not significantly affect the results.





For this case, this results in:

$$N_{\mathrm{DU,c}}^{\mathrm{imm(c)}} = N_{\mathrm{cm}} \tag{B18a}$$

$$N_{\mathrm{BC,c}}^{\mathrm{imm(c)}} = N_{\mathrm{glPOM,c}}^{\mathrm{imm(c)}} = N_{\mathrm{AmSu,c}}^{\mathrm{imm(c)}} = 0 \tag{B18b}$$

If $f_{\mathrm{DU}} < 0.7$, other INPs can play a major role and similar calculations as in the mixed accumulation mode are performed.

$$N_{\mathrm{glPOM,cm}}^{\mathrm{imm(c)}} = M_{\mathrm{glPOM,cm}} C_{\mathrm{glPOM,cm}} \tag{B19}$$

The number concentration of possible AmSu INPs in this mode is calculated from the fraction of the $\mathrm{AmSu_{mix,coa}}$ tracer in the mode.

$$N_{\mathrm{AmSu,cm}}^{(c)} = (1 - f_{\mathrm{cs}}) \cdot N_{\mathrm{AmSu_{mix,coa}}} \tag{B20}$$

Potential DU INP numbers are calculated from the DU mass in the mode and subsequently reduced by respective inclusions inside AmSu particles.

$$N_{\mathrm{DU,cm}}^{\mathrm{imm(c)}} = M_{\mathrm{DU,cm}} C_{\mathrm{DU,c}} - N_{\mathrm{incl}}^{\mathrm{DU}} \ , \tag{B21}$$

where

$$N_{\mathrm{incl}}^{\mathrm{DU}} = \frac{N_{\mathrm{DU,cm}}}{N_{\mathrm{cm}}} \cdot N_{\mathrm{AmSu,cm}}^{(c)} \ . \tag{B22}$$

This represents a minimum estimate of the number of DU particles in the mode, as also accumulation-mode dust may be present due to coagulation. Additionally the potential DU and AmSu INP number concentrations have to be reduced according
to glPOM numbers following the assumption of an organic shell around other particles (e.g. Smith et al., 2012, 2013; Schill et al., 2014; Saukko et al., 2015), by multiplication with the scaling factor

$$f_{\mathrm{glPOM}}^{\mathrm{cm}} = (N_{\mathrm{cm}} - N_{\mathrm{glPOM,cm}}^{(c)})/N_{\mathrm{cm}} \ . \tag{B23}$$

The remaining particles in the mode are available for immersion freezing of BC, as the possible contribution of accumulation-mode dust is probably small in this non-dust-dominated regime

$$N_{\mathrm{BC,cm}}^{\mathrm{imm(c)}} = \max(0, N_{\mathrm{cm}} - N_{\mathrm{DU,cm}}^{\mathrm{imm(c)}} - N_{\mathrm{glPOM,cm}}^{\mathrm{imm(c)}} - N_{\mathrm{AmSu,cm}}^{\mathrm{imm(c)}}) \ . \tag{B24}$$

The insoluble coarse mode is dominated by mineral dust, since coagulational growth of BC particles from the insoluble accumulation mode can be neglected (limited BC mass and low self-coagulation efficiency). Therefore, the number of available deposition freezing dust INPs is given by:

$$N_{\mathrm{DU,c}}^{\mathrm{dep(c)}} = N_{\mathrm{ci}} \tag{B25}$$



**Table B1.** Overview of the calculations of INP number concentrations per MADE3 mode and INP species. MADE3 Aitken, accumulation, and coarse modes are indicated with the indices $k$, $a$, and $c$, respectively. Mixing states are depicted by $s$, $i$, and $m$ for soluble, insoluble, and mixed, respectively. All calculated number concentrations undergo consistency checks in the code, to make sure that the estimated number concentrations in each mode are positive and do not exceed the total number concentration in the mode itself. A detailed description of these calculations is presented in the text.

| Mode | | DU | BCair | glPOM | AmSu | BC |
|---|---|---|---|---|---|---|
| km | | 0 | $M_{\mathrm{BCair,km}} C_{\mathrm{BCair,k}}$ | 0 | 0 | $N_{\mathrm{km}} - N_{\mathrm{BCair,k}}^{\mathrm{imm(c)}}$ |
| ki | | 0 | $M_{\mathrm{BCair,ki}} C_{\mathrm{BCair,k}}$ | 0 | 0 | $N_{\mathrm{ki}} - N_{\mathrm{BCair,k}}^{\mathrm{dep(c)}}$ |
| ks | | 0 | 0 | 0 | 0 | 0 |
| as | | 0 | 0 | $M_{\mathrm{glPOM,as}} C_{\mathrm{glPOM,as}}$ | $f_{\mathrm{as}} \cdot (N_{\mathrm{AmSu_{mixS}}} + N_{\mathrm{AmSu_{mixL}}}) - N_{\mathrm{glPOM,as}}^{(c)}$ | 0 |
| am | | $(M_{\mathrm{DU,am}} C_{\mathrm{DU,a}} - N_{\mathrm{incl}}^{\mathrm{DU}}) \cdot f_{\mathrm{glPOM}}^{\mathrm{am}}$ | $(M_{\mathrm{BCair,am}} C_{\mathrm{BCair,a}} - N_{\mathrm{incl}}^{\mathrm{BCair}}) \cdot f_{\mathrm{glPOM}}^{\mathrm{am}}$ | $M_{\mathrm{glPOM,am}} C_{\mathrm{glPOM,am}}$ | $(1 - f_{\mathrm{as}}) \cdot (N_{\mathrm{AmSu_{mixS}}} + N_{\mathrm{AmSu_{mixL}}}) \cdot f_{\mathrm{glPOM}}^{\mathrm{am}}$ | $N_{\mathrm{am}} - N_{\mathrm{DU,am}}^{\mathrm{imm(c)}} - N_{\mathrm{BCair,am}}^{\mathrm{imm(c)}} - N_{\mathrm{glPOM,am}}^{(c)} - N_{\mathrm{AmSu,am}}^{(c)}$ |
| ai | | $M_{\mathrm{DU,ai}} C_{\mathrm{DU,a}}$ | $M_{\mathrm{BCair,ai}} C_{\mathrm{BCair,a}}$ | 0 | 0 | $N_{\mathrm{ai}} - N_{\mathrm{DU,a}}^{\mathrm{dep(c)}} - N_{\mathrm{BCair,a}}^{\mathrm{dep(c)}}$ |
| cs | | 0 | 0 | $M_{\mathrm{glPOM,cs}} C_{\mathrm{glPOM,cs}}$ | $f_{\mathrm{cs}} \cdot N_{\mathrm{AmSu_{mix,coa}}} - N_{\mathrm{glPOM,cs}}^{(c)}$ | 0 |
| cm | $f_{\mathrm{DU}} \geq 0.7$ | $N_{\mathrm{cm}}$ | 0 | $M_{\mathrm{glPOM,cm}} C_{\mathrm{glPOM,cm}}$ | $(1 - f_{\mathrm{cs}}) \cdot N_{\mathrm{AmSu_{mix,coa}}} \cdot f_{\mathrm{glPOM}}^{\mathrm{cm}}$ | $N_{\mathrm{cm}} - N_{\mathrm{DU,cm}}^{\mathrm{imm(c)}} - N_{\mathrm{glPOM,cm}}^{\mathrm{imm(c)}} - N_{\mathrm{AmSu,cm}}^{(c)}$ |
| | $f_{\mathrm{DU}} < 0.7$ | $(M_{\mathrm{DU,cm}} C_{\mathrm{DU,c}} - N_{\mathrm{incl}}^{\mathrm{DU}}) \cdot f_{\mathrm{glPOM}}^{\mathrm{cm}}$ | | | | |
| ci | | $N_{\mathrm{ci}}$ | 0 | 0 | 0 | 0 |



**Table B2.** Overview of the parameters used to convert particle mass to number concentrations (using Eq. A6) per INP species and aerosol mode, i.e. particle diameter $D$, geometric standard deviation $\sigma_g$, and particle density $\rho$. MADE3 aerosol modes are abbreviated as in Table B1. For Amsu and BC particles, no mass-to-number conversion is necessary since AmSu is already a number density tracer and the BC number concentration is calculated as the remaining number in the respective mode.

| Aerosol mode | INP species | $D$ (µm) | $\sigma_g$ | $\rho$ (kg/m$^3$) | Reference |
|---|---|---|---|---|---|
| km, ki | BCair | 0.025 | 1.55 | 1500 | Petzold et al. (1999) |
| as | glPOM | 0.2 | 2.0 | 1000 | $D$, $\sigma$ similar to assumptions for AmSu particles (see Sect.3.3), $\rho$ according to the standard POM tracer (Kaiser et al., 2019) |
| am | glPOM | 0.191 | 2.0 | 1000 | 50 nm thick spherical shell around a 100 nm core, similar to assumed AmSu particle size (see Sect.3.3) |
| | BCair | 0.15 | 1.65 | 1500 | Petzold et al. (1999) |
| | DU | 0.42 | 1.59 | 2500 | Dentener et al. (2006) |
| ai | BCair | 0.15 | 1.65 | 1500 | Petzold et al. (1999) |
| | DU | 0.42 | 1.59 | 2500 | Dentener et al. (2006) |
| cs | glPOM | 2.0 | 2.2 | 1000 | $D$, $\sigma$ similar to assumptions for AmSu particles (see Sect.3.3), $\rho$ according to the standard POM tracer (Kaiser et al., 2019) |
| cm | glPOM | 1.91 | 2.2 | 1000 | 0.5 µm thick spherical shell around a 1 µm core, similar to assumed AmSu particle size (see Sect.3.3) |
| | DU | 1.3 | 2.0 | 2500 | Dentener et al. (2006) |

*Author contributions.* CB conceived the study, implemented the model developments concerning new types of INPs, designed and performed the model simulations, analysed the data, evaluated and interpreted the results, and wrote the paper. JH contributed to conceiving the study and to the model developments, the model evaluation, the interpretation of the results, and to the text. MR assisted in preparing the simulation setup, helped designing the evaluation methods, and contributed to the model developments, the interpretation of the results and to the text.

*Competing interests.* The authors declare that they have no conflict of interest.

*Acknowledgements.* The model simulations and data analysis for this work used the resources of the Deutsches Klimarechenzentrum (DKRZ) granted by its Scientific Steering Committee (WLA) under project ID bd0080. We are grateful to Daniel Sauer (DLR, Germany) for his comments and suggestions on an earlier version of the manuscript, and to George Craig (LMU, Germany), Sabine Brinkop, Patrick Jöckel, Christopher Kaiser, Robert Sausen and Helmut Ziereis (DLR, Germany) for helpful discussions. We are grateful for the support of the whole MESSy team of developers and maintainers.



This study was supported by the DLR transport programme (projects *Data and Model-based Solutions for the Transformation of Mobility – DATAMOST*, *Global model studies on the effects of transport-induced aerosols on ice clouds and climate, Transport and the Environment – VEU2*, and *Transport and Climate – TraK*), the DLR space research programme (projects *KliSAW* and *MABAK*), the German Federal Ministry for Economic Affairs and Climate Action – BMWK (project *Digitally optimized Engineering for Services – DoEfS; contract no. 20X1701B*), and the Initiative and Networking Fund of the Helmholtz Association (project *Advanced Earth System Modelling Capacity –*
*ESM*).





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
