# Peer review of "A global climatology of ice nucleating particles at cirrus conditions derived from model simulations with EMAC-MADE3"

_Atmospheric Chemistry and Physics, 2022_

## Author Comment (AC1)

**A global climatology of ice nucleating particles at cirrus conditions derived from model simulations with EMAC-MADE3 *C. Beer, J. Hendricks and M. Righi* Replies to referee comments**

We are grateful to the reviewers for their insightful comments and constructive criticism, which greatly helped us to improve the manuscript. Please find our point-by-point reply below (reviewers' comments are marked in blue, authors' reply in black, and text quotes in *"italic red"*).

Two major general changes have been included in the manuscript according to comments during the review process:

- Following the recommendations by the two reviewers, we changed the text in several parts bringing the large importance of ammonium sulfate ice nucleating particles more to the forefront of the manuscript.
- According to the editor's recommendations we shortened the manuscript by moving the Appendix sections to the Supplement.

**Reply to Reviewer #1 (Blaž Gasparini)**

Is there a way to find observational evidence of ammonium sulfate playing such a large role (outside of lab studies)? This would go against the generally accepted notion of dust as the key ice nucleating particle at cirrus levels (e.g. Cziczo et al., 2013, Froyd et al., 2022, and many more).

To the authors knowledge no specific atmospheric measurements focusing on the role of crystalline ammonium sulfate as ice nucleating particles exist. Future observational activities are necessary to improve the understanding of the ice nucleation by crystalline ammonium sulfate. We included some additional comments on this subject in the results section of the manuscript.

(line 424) "Especially the strong impact of crystalline ammonium sulfate INPs is remarkable, as several previous studies reported mineral dust as the single most important INP type (e.g. Cziczo et al., 2013; Froyd et al., 2022). This implies the need for dedicated measurements on the ice nucleation of ammonium sulfate in the cirrus regime. Additionally, future modelling efforts should take crystalline ammonium sulfate into account to further evaluate its importance for aerosol-induced cirrus formation."

What is the climatic role of different ice nucleating modes? Is the addition of ammonium sulfate as separate ice nucleating species as important also from the radiative perspective?

We agree with the referee that an evaluation of the radiative effects of the different INPs would be an interesting result. However, this is a quite extensive topic and, hence, would tend to go beyond the scope of this paper. A follow-up study is in preparation, which will analyse the radiative effects of INPs in detail. The results of a first set of simulations suggest a marked radiative effect due to crystalline ammonium sulfate ice nucleation, comparable with the effect of mineral dust and soot INPs.

Could you comment on the uncertainties in ammonium sulfate ice nucleating properties, e.g. in its onset ice nucleation temperature, critical supersaturation, etc? As an example, a recent study by Bertozzi et al., 2021 found the onset of ammonium sulfate freezing at -54°C at a critical supersaturation of 1.3. Such assumptions would probably significantly reduce the role of ammonium sulfate as ice nucleating particles compared to the used assumptions.

Indeed, the ice nucleating ability of crystalline ammonium sulfate is still uncertain. For example, organic coatings on ammonium sulfate particles could reduce their freezing potential (Ladino et al., 2014; Bertozzi et al., 2021). Reducing the activated fraction by one order of magnitude to fact=0.002 (for the comparison in Fig. 5), according to the results described in Bertozzi et al., 2021, would reduce the impact of ammonium sulfate INPs. However,

also in this case crystalline ammonium sulfate would still contribute a substantial fraction to the total INP number in large parts of the globe, e.g. with concentrations comparable to mineral dust in the Southern Hemisphere and at high northern latitudes (see Figure below, comparing the dust INP number concentration with the ammonium sulfate INP concentration assuming  $f_{act}$ =0.002).

We also added the following text to the manuscript:

(line 388) "Notably, the freezing efficiency of crystalline ammonium sulfate and glassy organic INPs is still uncertain, as only few laboratory studies investigated their ice nucleating abilities. A lower ice nucleating ability of crystalline ammonium sulfate, e.g. due to coatings of organic material (Ladino et al., 2014; Bertozzi et al., 2021), would reduce the impact of ammonium sulfate INPs. However, due to the large number of crystalline ammonium sulfate particles simulated here, they would probably still contribute a substantial fraction to the total INP number in large parts of the globe."

Detrainment of ice is one of the key sources of upper tropospheric clouds and likely accounts for most of the upper tropospheric clouds in the tropics below about 14 km and a large fraction of the extratropical summertime high clouds. How does detrainment affect in-situ cirrus and their ice nucleation? How are the ice number and mass sources of detrainment treated, and does detrainment interact with in-situ ice nucleation?

Besides homogeneous freezing and the different heterogeneous freezing modes, the model also considers preexisting ice crystals. This pre-existing ice mode represents all ice crystals which are already present in a grid box before the calculations of in-situ aerosol-induced freezing start. This pre-existing mode comprises, besides ice crystals from previous time steps, also ice crystals transported into cirrus clouds but originating from other sources such as detrainment from deep convective clouds. In this way, the pre-existing ice crystals from detrainment compete with the other freezing pathways, i.e. heterogeneous and homogeneous freezing (Righi et al., 2020).

Page 1, line 2: the word climate modifications alludes to artificial geoengineering-type of modifications. You could consider using a synonym.

Thank you for pointing this out. We changed the text accordingly: "... climate effects"

Page 1: The last sentence of the abstract describes (in my opinion) the main result of the manuscript. The reader should ideally be informed of the key result earlier than at the end of a relatively long abstract.

Thank you for the suggestion. We shortened and changed the abstract accordingly, focusing more on the main results about glassy organics and crystalline ammonium sulfate.

Page 7: It seems like you have put a lot of effort into numerically representing and due to the need for 3 additional tracers also computing ammonium sulfate aerosols. Is there a simpler way of simulating crystalline ammonium sulfate that the other modeling groups may be more likely to implement in their models?

The implementation of ammonium sulfate INPs is not trivial, as the phase transition has to be considered to simulate crystalline ammonium sulfate particles. Nonetheless, some simplifications compared to the implementation described here are conceivable:

- Considering only one solid ammonium sulfate tracer with a characteristic accumulation mode size
- Assuming only externally mixed ammonium sulfate particles (without possible mixtures with other aerosol species)
- Assuming a characteristic value for the efflorescence relative humidity instead of calculating ERH depending on the ammonium-to-sulfate ratio and possible interactions with mineral dust particles

**We also added the following sentence to the text:**

(line 298) "Therefore, considering only one externally mixed ammonium sulfate tracer could be a reasonable simplification to include crystalline ammonium sulfate also in other model systems to further elucidate its impacts."

**Figure 1, panel (a): Surface or 300 hPa level?**

Thanks for spotting this. This was intended to depict the 300 hPa level; the figure has been corrected accordingly.

Figure 5: panel (c) should use the same colorbar limits as all other panels, not to artificially overemphasize the role of aircraft soot.

We tried using the same color scale as in the other panels, however, this would lead to an almost invisible signal from BCair due to the low number concentrations. Therefore, we chose a different color scale, but now also state this in the panel heading (in addition to the figure caption).

Page 19, lines 414-117: The two sentences seem to be contradicting each other (the first one claiming the occurrence frequencies of heterogeneous freezing are largely preventing homogeneous freezing, the second one saying that heterogeneous INPs do not completely suppress homogeneous freezing)

**Thank you for spotting this inconsistency, we rephrased this sentence as follows:**

" (line 414) [...] indicating that heterogeneous freezing often prevents homogeneous ice nucleation. However, in those cases where homogeneous freezing takes place, it produces very large concentrations of ice crystals."

Page 19, lines 429 and further: I agree INP measurements at cirrus conditions are practically non-existent, but you may still want to comment on the surprising result that ammonium sulfate is found to be the key INP species at cirrus conditions, contrary to the mainstream cirrus literature.

Thank you for bringing this up, we added the following to the text:

(line 424) "Especially the strong impact of crystalline ammonium sulfate INPs is remarkable, as several previous studies reported mineral dust as the single most important INP type (e.g. Cziczo et al., 2013; Froyd et al., 2022). This implies the need for dedicated measurements on the ice nucleation of ammonium sulfate in the cirrus regime. Additionally, future modelling efforts should take crystalline ammonium sulfate into account to further evaluate its importance for aerosol-induced cirrus formation."

Page 21, lines 477-484: You don't comment at all about the surprising result of the large importance of ammonium sulfate! From this paragraph, it sounds like dust is clearly the main INP, which is different from your model results.

**We added the following sentence to the paragraph:**

(line 479) "Importantly, crystalline ammonium sulfate INP concentrations are comparable or even exceed those of mineral dust in large parts of the globe, e.g. the Southern Hemisphere and the high northern latitudes."

Page 22, lines 497-500: I would suggest modifying for clarity the italicized part of the sentence: "....this study demonstrates the importance of including *additional ice nucleating particle types* together with.... " to ammonium sulfate. And then just conclude with "Glassy organic particles probably have only a minor influence..."

**Thank you. We changed the text as suggested:**

(line 499) "To conclude, the climatology of ice nucleating particles at cirrus formation presented in this study demonstrates the importance of including crystalline ammonium sulfate together with mineral dust and soot particles in global models. Glassy organic particles probably have only minor influences, as their INP concentrations in the upper troposphere are mostly small. The remarkable large importance of crystalline ammonium sulfate shown here should be further investigated in dedicated observation campaigns and modelling efforts."

**Reply to anonymous Reviewer #2**

(1) The two major outcomes of the study are not really prominently stated. It was claimed for a long time that glassy organic aerosols are an important class of INPs and they must have an impact. It is quite clear from this study that the impact is rather weak, if not negligible. The dominant role of ammonium sulfate is also new. I would suggest to emphasize these interesting results more prominently.

**Thank you for pointing this out. We changed the manuscript in several places to address this (see also the replies to Reviewer #1), e.g.**

(line 499) "To conclude, the climatology of ice nucleating particles at cirrus formation presented in this study demonstrates the importance of including crystalline ammonium sulfate together with mineral dust and soot particles in global models. Glassy organic particles probably have only minor influences, as their INP concentrations in the upper troposphere are mostly small. The remarkable large importance of crystalline ammonium sulfate shown here should be further investigated in dedicated observation campaigns and modelling efforts."

(2) For aerosols, the large scale transport is the most important pathway, thus the distribution of aerosols can be simulated well with GCMs, even in this coarse resolution of T63. However, for treating (ice) clouds in GCMs, the variability of thermodynamic variables plays an important role. The ice cloud model relies on former work for the EMAC model, which only marginally treats subgrid scale variability, e.g., using TKE or gravity wave drags for determining the variability of vertical velocity; however, the horizontal/vertical variability is only taken into account by crude cloud cover schemes. From observations (satellite, surface observations, and many others) we know quite well that also ice clouds have internal structures, leading to heterogeneous cloud layers, which provide additional spatial and temporal time scales. The use of microphysics schemes for the whole grid box, driven by a large-scale motion, without these scales in between might lead to an overestimation of the impact of nucleation pathways. A similar issue might occur for the removing of INPs (as included in ice crystals) by sedimentation of ice crystals, since sedimentation of ice particles in quite coarse vertical resolutions is highly tuned. It is quite clear that the authors cannot change the coupled ice cloud scheme. However, I would suggest to add some comments on these issues, since the competition of heterogeneous and homogeneous nucleation might be affected by small/meso scale motions.

Thank you for bringing this up. We included some comments on these issues of uncertainties to the conclusion section of the manuscript.

(line 504) "The described results concerning INPs and their interaction with cirrus clouds are still subject to uncertainties. Most notably, the cloud microphysics scheme mainly relies on calculations for the whole model grid box. Subgrid-scale processes are parametrized, e.g. the variability in the vertical velocity or the sedimentation of ice crystals (including the removal of embedded INPs; Kuebbeler et al., 2014; Righi et al., 2020). However, these parametrizations are limited and introduce additional uncertainties, which could lead to a misrepresentation of the impact of heterogeneous ice nucleation pathways. Additionally, the presented results are dependent on the freezing efficiency of the different INPs. For instance, Righi et al. (2021) employing a similar model setup found a strong dependency of the aviation soot–cirrus effect on the assumed ice nucleating properties of soot particles. This uncertainty could be further analysed in future studies by varying the ice nucleation thresholds of the different INPs." (3) How sensitive are these results if the nucleation thresholds for the different INPs are changed (within their uncertainties)? Have you checked this in sensitivity analyses?

The results are indeed dependent on the freezing properties of the different INPs. For example, Righi et al. (2021) demonstrated a strong relationship between the ice nucleating properties of aviation soot and the resulting radiative effect. We added a corresponding comment about the uncertainty concerning this aspect (see the text below). A dedicated sensitivity study on this aspect is planned for future analyses but could not be realized here, as performing a large number of simulations with the current setup (T63) would be computationally too expensive. For such a sensitivity study a lower model resolution will have to be employed.

(line 508) Additionally, the presented results are dependent on the freezing efficiency of the different INPs. For instance, Righi et al. (2021) employing a similar model setup found a strong dependency of the aviation sootcirrus effect on the assumed ice nucleating properties of soot particles. This uncertainty could be further analysed in future studies by varying the ice nucleation thresholds of the different INPs."